# A TRIM71 binding long noncoding RNA Trincr1 represses FGF/ERK signaling in embryonic stem cells

Ya-Pu Li[1], Fei-Fei Duan[1], Yu-Ting Zhao[1], Kai-Li Gu[1], Le-Qi Liao[1], Huai-Bin Su[1], Jing Hao[1], Kun Zhang[2], Na Yang[2] & Yangming Wang [1]

Long noncoding RNAs (lncRNAs) have emerged as important components of gene regulatory network in embryonic stem cells (ESCs). However, the function and molecular mechanism of lncRNAs are still largely unknown. Here we identifies Trincr1 (TRIM71 interacting long noncoding RNA 1) lncRNA that regulates the FGF/ERK signaling and self-renewal of ESCs. Trincr1 is exported by THOC complex to cytoplasm where it binds and represses TRIM71, leading to the downregulation of SHCBP1 protein. Knocking out Trincr1 leads to the upregulation of phosphorylated ERK and ERK pathway target genes and the decrease of ESC self-renewal, while knocking down Trim71 completely rescues the defects of Trincr1 knockout. Furthermore, ectopic expression of Trincr1 represses FGF/ERK signaling and the self-renewal of neural progenitor cells (NPCs). Together, this study highlights lncRNA as an important player in cell signaling network to coordinate cell fate specification.

[1] Beijing Key Laboratory of Cardiometabolic Molecular Medicine, Institute of Molecular Medicine, Peking University, 100871 Beijing, China. [2] State Key Laboratory of Medicinal Chemical Biology, College of Pharmacy, Nankai University, 300353 Tianjin, China. These authors contributed equally: Ya-Pu Li, Fei-Fei Duan. Correspondence and requests for materials should be addressed to Y.W. (email: yangming.wang@pku.edu.cn)

Mammalian genomes are pervasively transcribed to generate thousands of lncRNAs. The majority of lncRNAs are restricted to specific cell lineages and developmental stages[1,2], suggesting regulatory roles in cell fate specification and determination. Recently, dozens of lncRNAs are identified as potential regulators in the self-renewal and differentiation of ESCs[3,4]. Until today, their regulatory mechanisms are still largely unknown. Key mechanisms controlling ESC self-renewal and differentiation include signaling pathways, epigenetic, transcriptional, and post-transcriptional regulations. The association of many lncRNAs with chromatin or epigenetic regulatory complexes indicates their roles in epigenetic and transcriptional regulations[4]. Nevertheless, despite the importance of various signaling pathways in ESCs, the regulation of signaling pathways by ESC-enriched lncRNAs has never been reported.

Fibroblast growth factors (FGF)/extracellular signal regulated kinase (ERK) signaling controls a multitude of cell fate choices including self-renewal and differentiation[5,6]. Cellular responses to FGF/ERK signaling are extremely complicated and highly cell type specific. FGF/ERK pathway is essential for the survival and proliferation of many differentiated cells, as well as mouse epiblast stem cells and human ESCs[7]. However, its activity has to be effectively inhibited in mouse ESCs, because the activation of FGF/ERK signaling drives ESCs out of naive pluripotency state to enter into primed pluripotency state, which is then moderated by a variety of lineage differentiation cues[8]. Unexpectedly, pluripotency transcription factors Oct4 and Sox2 promote the expression of autocrine FGF4, which acts through its cognate receptor FGFR2 to induce potent ERK signaling[9,10]. The auto-inductive FGF4/ERK signaling has to be controlled below a certain threshold for optimal self-renewal and pluripotency in ESCs. Conversely, during differentiation, FGF/ERK signaling must be efficiently activated to allow ESCs entering into a state responsive to lineage inducing factors. Therefore, fine-tuning the choice and amplitude during the relay of FGFR-ERK signaling is essential for appropriate ESC self-renewal and differentiation.

FGF signaling is initiated through the auto-phosphorylation of intracellular tyrosine residues in an FGF receptor (FGFR) that is induced upon a ligand-receptor interaction[5,6]. Activated FGFR phosphorylates docking proteins such as FRS2 and SHC[11], which further recruits GRB2-SOS complex. The relocation of SOS on cell membrane induces RAS activation which eventually leads to the activation and nuclear translocation of ERK through a RAF/MEK/ERK kinase cascade[5,6]. The FGF/ERK signaling can be regulated by protein levels of core members of signal transduction pathways and their associated proteins which negatively or positively modulate the signaling relay through physical interaction or enzymatic activities (e.g., DUSP)[12]. Recently, multiple factors such as microRNAs (miRNAs)[13], the number of active X chromosomes[14] as well as PRC2 components[15] have been reported to regulate the activity of ERK pathway in ESCs with unidentified mechanisms. Besides these reports, the regulation of FGF/ERK signaling in ESCs is largely unexplored. *Trim71*, also known as *lin-41*, was first discovered as a heterochronic gene controlling cell fate decision of seam cells in *C. elegans*[16]. It also play important functions during neural tube closure and postnatal development in mouse[17,18] and promote the reprogramming of human pluripotent cells[19]. These functions are often attributed to its mRNA suppressing function through directly binding mRNA[20]. Interestingly, TRIM71 is recently found to promote FGF/ERK signaling pathway by binding and stabilizing SHCBP1 in neural progenitor cells[21]. However, whether TRIM71 regulates FGF/ERK signaling and how its activity is regulated in ESCs are not known.

In this study, we confirm that TRIM71 and SHCBP1 promote the activity of FGF/ERK pathway in ESCs. More importantly, we identify a TRIM71 associated lncRNA Trincr1 that fine tunely represses FGF/ERK signaling through suppressing TRIM71, therefore positively regulating the self-renewal of ESCs. We show that the cytoplasmic localization of Trincr1 is dependent on THOC5. Finally, we demonstrate that the ectopic expression of Trincr1 also inhibits FGF/ERK signaling in NPCs, in turn negatively impacting the self-renewal of NPCs. Together, our study elucidates important function of a lncRNA in regulating FGF/ERK signaling and self-renewal of ESCs.

## Results

**Trim71 promotes FGF/ERK signaling in ESCs**. We first tested whether Trim71 plays a conserved function in FGF/ERK signaling in ESCs as in NPCs[21]. Since serum may interfere with the study of signaling pathways, we performed experiments in ESCs cultured in chemically defined media supplemented with MEK inhibitor PD0325901 (PD), GSK3 inhibitor CHIR99021 (CHIR), and leukemia inhibitory factor (2i + LIF)[22]. We designed shRNAs and siRNAs to knockdown Trim71 and found that knocking down Trim71 significantly repressed FGF induced ERK phosphorylation in V6.5 ESCs (Supplementary Fig. 1a–d). We noticed that Trim71 knocking down ESCs grew slightly slower than control ESCs (Supplementary Fig. 1e), consistent with a previous study[23]. Additionally, Trim71 shRNA ESCs expressed lower levels of SHCBP1 protein (Supplementary Fig. 1b), consistent with the finding from NPCs[21]. To further verify the function of Trim71 and Shcbp1 in FGF/ERK signaling in ESCs, we designed two gRNAs for each gene to knockdown Trim71 and Shcbp1 using CRISPRi strategy[24] and found that knocking down either Trim71 or Shcbp1 could repress FGF induced ERK phosphorylation in ESCs (Supplementary Fig. 1f–h). In addition, although SHCBP1 protein level was decreased around 50–60% in Trim71 knocking down ESCs and almost completely abolished in Shcbp1 knocking down ESCs (Supplementary Fig. 1g), FGF/ERK signaling was similarly repressed in Trim71 and Shcbp1 CRISPRi ESCs (Supplementary Fig. 1h), suggesting that TRIM71 may promote FGF/ERK signaling only partially through SHCBP1. Finally, consistent with their positive regulatory roles in FGF/ERK signaling, knocking down Trim71 and Shcbp1 significantly increased colony formation ability of ESCs in N2B27 + LIF condition (Fig. 1a). Together, these data show that similar to its role in neural progenitor cells, Trim71 promotes FGF/ERK signaling in ESCs and its function is at least partially through upregulating SHCBP1 protein.

**Trim71 binds a lincRNA Trincr1 in ESCs**. The enrichment of Trim71 in ESCs and its function in promoting FGF/ERK signaling suggest the existence of other factors restraining its signaling promoting activity. We decided to focus on TRIM71 interacting RNAs, particularly noncoding RNAs. To identify RNAs bound by TRIM71, we performed RNA immunoprecipitation and sequencing (RIP-Seq) in ESCs overexpressing FLAG-TRIM71 using FLAG antibody, since we were not able to obtain a high quality TRIM71 antibody. A total of 495 transcripts were identified as bound by FLAG-TRIM71 using > 3 fold enrichment and FPKM value > 2 in input as a cutoff (Fig. 1b and Supplementary Data 1). More excitingly, we found three long intergenic noncoding RNAs (lincRNAs) bound by TRIM71. Among them, Gm2694 was the most enriched lincRNA in IP sample versus input (Fold of enrichment ~70) (Fig. 1b, c and Supplementary Data 1). In addition, 8 out of 9 candidate TRIM71-binding RNAs including Gm2694 based on RIP-seq data were confirmed by RIP-qPCR (Fig. 1d and Supplementary Fig. 1i, 1j). Next, we evaluated the coding potential of Gm2694 using Coding Potential Calculator and CPAT, and both programs independently

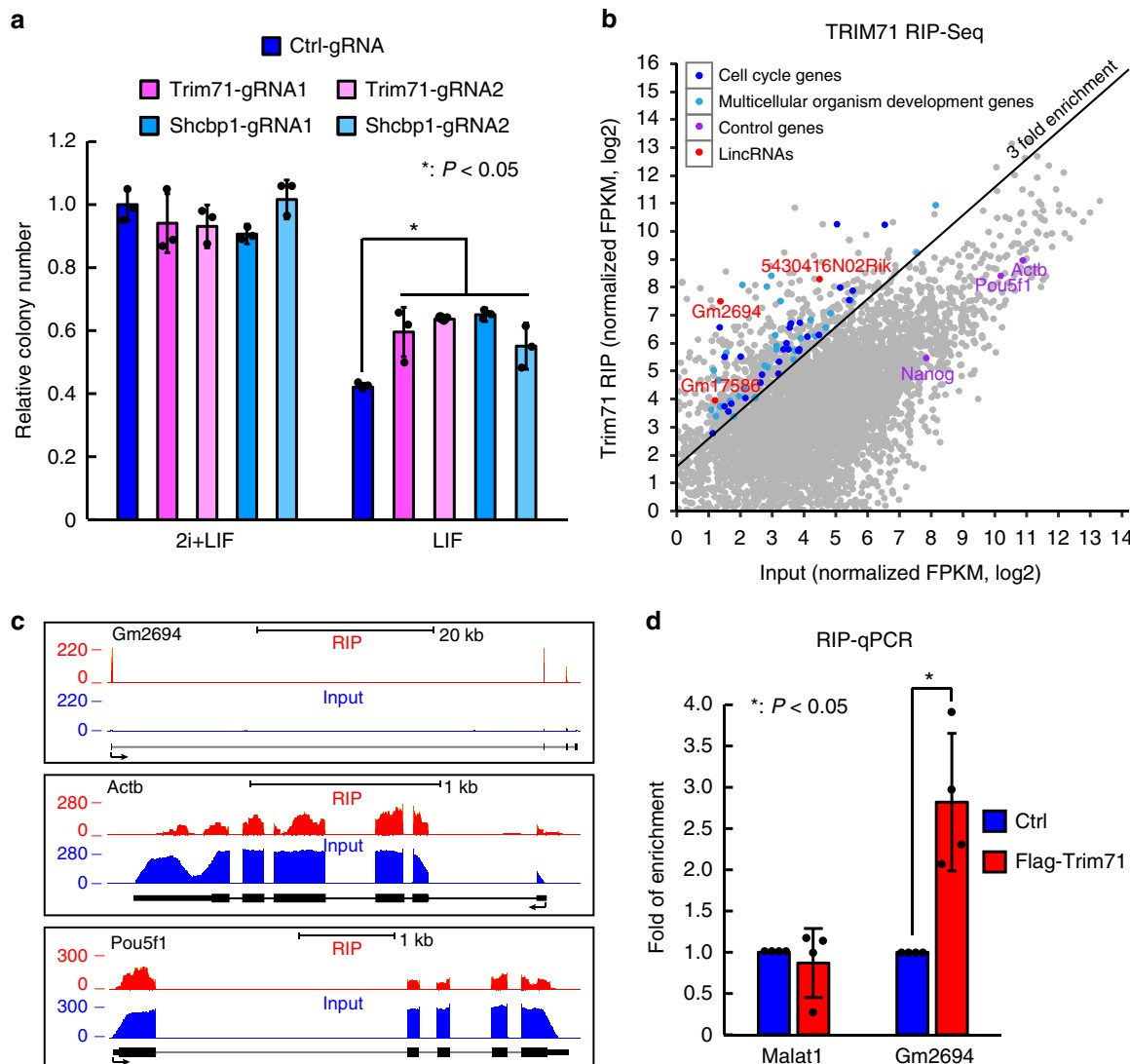

**Fig. 1** Trim71 promotes FGF/ERK signaling and binds lncRNA Trincr1 in ESCs. **a** Colony formation assay for ESCs treated with control, Trim71, and Shcbp1 CRISPRi gRNAs. For the LIF condition, cells were plated in 2i + LIF medium for 1 day, then changed to N2B27 supplemented with LIF for 4 days before being re-plated in 2i + LIF medium for colony formation. Data were normalized to ESCs treated with control gRNAs grown in 2i + LIF. $n = 3$ biological replicates. **b** RIP-Seq analysis showing average-normalized, log2 transformed FPKM value of Flag-Trim71-RIP and input. **c** RIP-Seq tracks at the Gm2694, Actb, and Oct4 (also known as Pou5f1) loci. Shown are normalized read counts per million. Flag-Trim71-RIP and input were scaled to the same level. **d** RIP followed by RT-qPCR analysis of Malat1 and Gm2694. Data were normalized to ESCs transfected with empty 3XFLAG overexpression vectors (Ctrl). $n = 4$ independent experiments. Shown are mean ± SD. For **a**, $P$ values were determined by unpaired two-way ANOVA with two-sided Dunnett's test. For **d**, $P$ values were determined by paired two-sided Student's $t$-test

predicted that Gm2694 is a noncoding RNA[25,26]. Moreover, Ribosome-seq analysis in mouse ESCs shows that Gm2694 is not translated[27,28]. From this point, Gm2694 is renamed as Trincr1 for <u>TR</u>IM71 <u>i</u>nteracting <u>n</u>on<u>c</u>oding <u>R</u>NA1.

**Trincr1 represses FGF/ERK signaling in ESCs**. Next, we characterized molecular details of Trincr1 in ESCs. RACE (<u>r</u>apid <u>a</u>mplification of <u>c</u>DNA <u>e</u>nds) cloning followed by Sanger sequencing identified two Trincr1 isoforms sharing exactly the same last three exons (exon 2–4) but with different transcription start sites (Fig. 2a). The long isoform (Trincr1_L) has an extra 62 bases at the 5′end of the first exon comparing to the short isoform (Trincr1_S). A modified RT-qPCR assay showed that around 60 and 40% of Trincr1 are expressed as short and long isoforms, respectively (Fig. 2b and Supplementary Fig. 2a, b). Furthermore,

Trincr1 was significantly downregulated while Trim71 remained unchanged during ESC differentiation (Fig. 2c).

Next, we tested the role of Trincr1 in FGF/ERK signaling. Knocking down Trincr1 by two shRNAs caused substantial increases of basal ERK phosphorylation level (Supplementary Fig. 2c), suggesting a repressive role of Trincr1 in FGF/ERK signaling. To further confirm its function in FGF/ERK signaling and exclude off-target effects of shRNAs, we constructed Trincr1 knockout ESCs. With two guide RNAs targeting the flanking sequence of exons 2–4 (Fig. 2a), we successfully generated Trincr1−/− ESCs (Fig. 2d). Trincr1−/− ESCs proliferated at a similar rate as wild-type ESCs (Supplementary Fig. 2d). Consistent with results from Trincr1 shRNA ESCs, the basal ERK phosphorylation level was increased around 1.6 fold and 2.4 fold in Trincr1−/− ESCs in 2i + LIF and PD + LIF condition (Fig. 2e), respectively. More importantly, overexpressing either Trincr1_L or Trincr1_S in ESCs

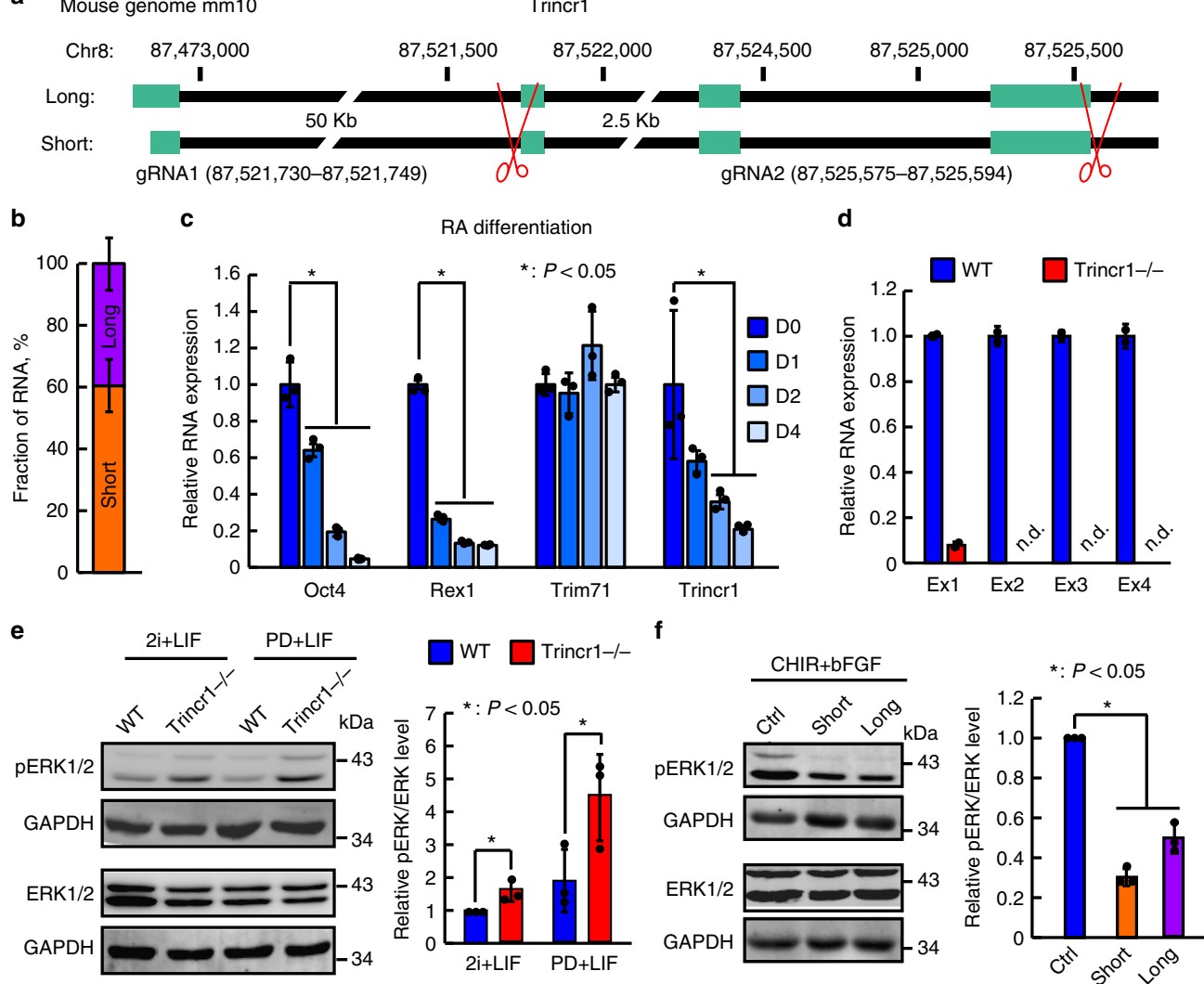

**Fig. 2** Trincr1 represses FGF/ERK signaling in ESCs. **a** Gene structure of Trincr1_L and S. Exon 1 (L): 87,472,809–87,472,960; Exon 1 (S): 87,472,871–87,472,960; Exon 2: 87,521,736–87,521,813; Exon 3: 87,524,296–87,524,427; Exon 4: 87,525,415–87,525,554; Cleavage and polyadenylation signal sequence "AAUAAA": 87,525,535–87,525,540. Scissors indicate the position of two gRNA sequences used to knockout Trincr1 in this study. **b** Fraction of Trincr1_L and S in ESCs. **c** RT-qPCR analysis of Trincr1 and Trim71 in differentiated ESCs induced by 100 nM All-Trans-Retinoic Acid for 4 days. The Gapdh gene was used as a control. For each gene, data were normalized to the RNA level of undifferentiated ESCs (day 0). $n = 3$ biological replicates. Appropriate differentiation is supported by the expression change of Oct4 and Rex1. **d** RT-qPCR analysis of Trincr1 in ESCs with Trincr1 knocked out by Cas9. Four sets of primers corresponding to four exons were used to confirm the loss of Trincr1 in *Trincr1−/−* ESCs. The β-actin gene was used as a control. Data were normalized to wild-type ESCs. $n = 2$ biological replicates. **e** Western blotting analysis of phosphorylated ERK in wild type and *Trincr1−/−* ESCs in 2i + LIF and PD + LIF. For quantification of pERK/ERK, data were normalized to GAPDH and then to wild-type ESCs cultured in 2i + LIF. $n = 3$ independent experiments. **f** Western blotting analysis of phosphorylated ERK in control and Trincr1 overexpressing ESCs induced by bFGF. For quantification of pERK/ERK, data were normalized to GAPDH and then to control overexpression vector transfected ESCs cultured in CHIR and induced by 12 ng per ml bFGF. $n = 3$ independent experiments. Shown are mean ± SD. For **c** and **f**, $P$ values were determined by unpaired two-way and one-way ANOVA with two-sided Dunnett's test, respectively. For **e**, $P$ values were determined by paired two-sided Student's $t$-test

significantly suppressed the phosphorylation of ERK upon induction by bFGF (Fig. 2f, and Supplementary Fig. 2e, f). Together, these data demonstrate that both isoforms of Trincr1 can repress FGF/ERK signaling in ESCs.

**ESC self-renewal requires Trincr1 in sub-optimal conditions.** Mouse ESCs are optimally maintained in chemically defined media with inhibitors to MEK and GSK3 and leukemia inhibitory factor (2i + LIF)[22,29]. In addition, any two components of 2i and LIF is sufficient but not optimal to maintain the self-renewal and

pluripotency state of ESCs[8,29]. We tested different media with different combinations of 2i and LIF and analyzed the expression of Oct4, Nanog, and Rex1 (also known as Zfp42) in *Trincr1−/−* versus wild-type ESCs (Supplementary Fig. 3a). The results from these experiments revealed that the loss of Trincr1 diminished the expression of pluripotency genes in PD + LIF, PD, LIF, or N2B27 medium without any supplemental factors. We then focused our further analysis in PD + LIF condition since wild-type ESCs essentially maintain their self-renewal ability under this condition. RT-qPCR analysis for a panel of pluripotency markers and early differentiation marker Fgf5 confirmed that Trincr1 is

important for the maintenance of ESC identity in PD + LIF (Fig. 3a). Colony formation assay showed that the ability to form a colony at single cell level was significantly decreased in two independent *Trincr1*−/− ESC lines in PD + LIF but not 2i + LIF condition (Fig. 3b). In addition, the loss of Trincr1 did not affect the expression of pluripotency genes in ESCs grown in conventional serum + LIF culture, but accelerated the differentiation upon LIF withdrawn (Supplementary Fig. 3b–d). Together, these data demonstrate that Trincr1 is important for the self-renewal of ESCs cultured in various sub-optimal conditions.

**Trincr1 represses target genes of ERK pathway in ESCs.** To further analyze the impact of Trincr1 knockout on the transcriptome of ESCs, we performed RNA-Seq for wild type and *Trincr1*−/− ESCs in both 2i + LIF and PD + LIF conditions (Supplementary Data 2). Principal component analysis showed that wild type and *Trincr1*−/− ESCs were clustered closely together in 2i + LIF, but were clearly separated away from each other in PD + LIF (Fig. 3c and Supplementary Fig. 3e). These results suggest that 2i + LIF mask the functional importance of Trincr1. The results for *Trincr1*−/− in 2i + LIF condition are not surprising, since 2i + LIF may stimulate multiple redundant pathways to support the self-renewal of ESCs. For example, ESCs lacking key transcription factors such as Klf2[30] and Esrrb[31] can still self-renew in 2i+LIF.

To get a more thorough view on the impact of *Trincr1* knockout, we performed KEGG pathway analysis for genes with average FPKM ≥ 1 and up- or downregulated >2 fold in *Trincr1*−/− versus wild-type ESCs in PD + LIF. Interestingly, genes upregulated were significantly enriched in several signaling pathways including MAPK signaling pathway, Wnt signaling pathway, and signaling pathways regulating pluripotency of stem cells (Fig. 3d, Supplementary Fig. 3f). Furthermore, we performed detailed analysis on four known pluripotency regulating signaling pathways[14], including AKT, MEK/ERK, STAT3, and GSK3 pathways. Among them the target genes of MEK/ERK pathway were the most significantly upregulated (~1.4 fold in median) in *Trincr1*−/− versus wild-type ESCs in PD + LIF (Supplementary Fig. 4), consistent with the increase of basal ERK phosphorylation level in *Trincr1*−/− ESCs. Gene set enrichment analysis (GSEA)[32] verified that MEK/ERK targets were strongly associated with *Trincr1*−/− versus wild-type ESCs in PD + LIF (Fig. 3e). Therefore, these data show that the repression of ERK phosphorylation by Trincr1 has functional consequences on the expression of downstream target genes and cell fate.

**Thoc5 regulates the export of Trincr1 to cytoplasm.** The interaction between Trincr1 and TRIM71 suggests the cytoplasmic presence of Trincr1. To test this, we determined cytoplasmic/nuclear distribution of Trincr1 in wild-type ESCs. Subcellular fractionation assay followed by RT-qPCR revealed that around 70% of Trincr1 is located in the cytoplasm (Fig. 4a). Previous report showed that Thoc5 is essential for the nuclear export of a subset of pluripotency regulating mRNAs[33]. Since Trincr1 promotes self-renewal and pluripotency of ESCs, we tested whether Thoc5 is also responsible for the nuclear export of Trincr1. We knocked down various components of transcription-export complex including Thoc5, Thoc2, Nxf1, and Aly (Fig. 4b) using a CRISPRi strategy. Subcellular fractionation assay showed that only knocking down Thoc5 inhibited the cytoplasmic transport of Trincr1 (Fig. 4c). Consistent with Trincr1 functioning in the cytoplasm, ERK phosphorylation was significantly upregulated in Thoc5 knocking down ESCs (Fig. 4d). These results show that Trincr1 is exported by Thoc5 to interact with TRIM71 in the cytoplasm.

**Sequence requirements for interaction of Trincr1 and TRIM71.** Next, we tried to determine which part of Trincr1 is responsible for inhibiting ERK phosphorylation. We checked ERK phosphorylation level upon FGF induction in ESCs overexpressing truncated Trincr1 (Fig. 5a, b and Supplementary Fig. 5a, b). The results show that the core functional sequence of Trincr1 existed in the 1–140th nt fragment of Trincr1_S (or 63rd-202nd nt fragment of Trincr1_L). This is also another evidence suggesting that Trincr1 functions as a noncoding RNA since the longest open reading frame found in this fragment codes for a peptide with only 6 amino acids[25]. These data suggest that an essential RNA fragment around 140 nt long near the 5′ end of Trincr1 functionally inhibits the phosphorylation of ERK induced by bFGF in ESCs.

To confirm that Trincr1 directly interacts with TRIM71, we performed RNA pull-down followed by mass spectrometry analysis to identify Trincr1 interacting proteins (Fig. 5c). T3 and T5 were synthesized by in vitro transcription and used as a bait and a control, respectively (Supplementary Fig. 5c). Consistently, endogenous TRIM71 was reproducibly identified with eight other proteins in samples pulled down by T3 from two independent experiments (Fig. 5d). RNA pull-down followed with western blotting analysis confirmed that TRIM71 was specifically bound by T3, but not by antisense T3 or a 240 nt fragment of lncRNA Malat1 (Fig. 5e). TRIM71 contains a TRIM motif consisting of a RING, two B-boxes and a CC domain, followed by filamin homology and NHL domains (Fig. 5f)[34]. Truncation experiments indicated that the NHL domain but not the RING or B1 box domain of TRIM71 was required for the binding of Trincr1 (Fig. 5g and Supplementary Fig. 5d). RIP-qPCR analysis further confirmed the interaction between Trincr1 and TRIM71 in ESCs and that the interaction is dependent on the NHL domain but not the RING and B1 box domain (Fig. 5h and Supplementary Fig. 5e). These data show that the physical interaction between Trincr1 and TRIM71 relies on the NHL domain of TRIM71 and the 5′ half of Trincr1.

**Trincr1 functions through suppressing TRIM71.** Next, we tested whether Trincr1 functions through direct inhibition of TRIM71 by epistasis analysis. First, we checked whether knocking down Trim71 can rescue the defects of *Trincr1*−/− in FGF/ERK signaling and ESC self-renewal. We knocked down Trim71 using shRNAs in *Trincr1*−/− ESCs (Fig. 6a) and found that knocking down Trim71 successfully rescued the basal level of ERK phosphorylation in *Trincr1*−/− ESCs in 2i + LIF condition (Fig. 6b, c) and serum + LIF condition (Supplementary Fig. 6a). In addition, no further inhibition of ERK phosphorylation was observed upon knocking down Trim71 in Trincr1_S overexpressing ESCs (Supplementary Fig. 6b), suggesting that Trim71 and Trincr1 function in the same pathway. In agreement with ERK phosphorylation analysis, colony formation assay showed that knocking down Trim71 in *Trincr1*−/− background fully rescued the self-renewal defect of *Trincr1*−/− ESCs in PD + LIF (Fig. 6d). Interestingly, although loss of Trincr1 did not significantly affect the proliferation of ESCs, it sensitized ESCs to depend on Trim71 for rapid proliferation (Fig. 6e), suggesting that Trim71 and Trincr1 cooperate to ensure the robustness of regulatory networks controlling ESC proliferation. To provide further mechanistic insights on Trincr1 function in FGF/ERK signaling, we checked the impact of Trincr1 knockout on the protein level of SHCBP1, the direct target of TRIM71 in regulating ERK pathway[21]. Compared to wild-type ESCs, SHCBP1 protein but not mRNA was significantly upregulated in *Trincr1*−/− ESCs (Fig. 6f). The increase in SHCBP1 protein was not due to the change in

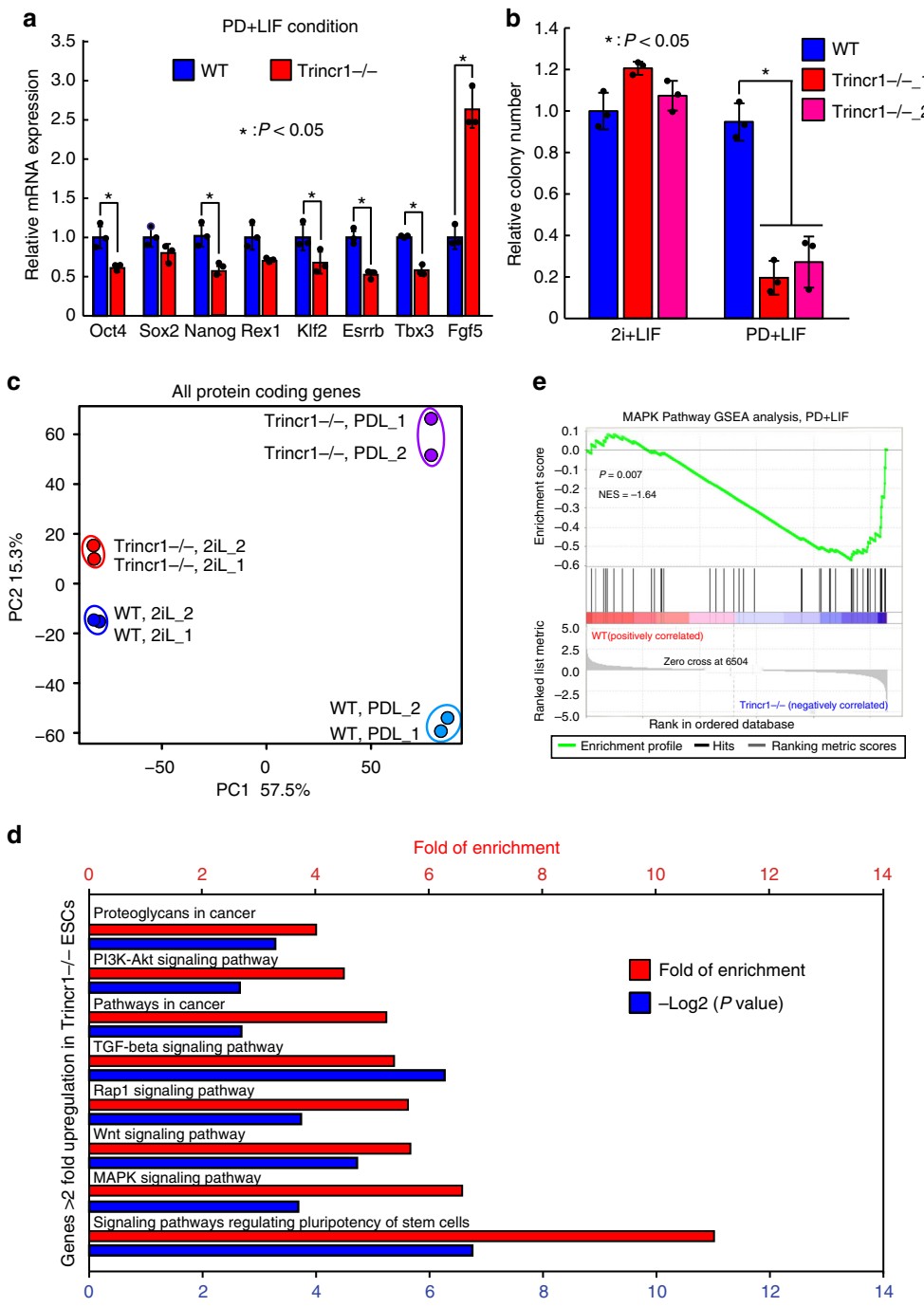

**Fig. 3** Trincr1 promotes ESC self-renewal and suppresses ERK target genes. **a** RT-qPCR analysis of pluripotency genes and Fgf5 in *Trincr1−/−* ESCs in PD + LIF condition. The β-actin gene was used as a control. For each gene, data were normalized to the mRNA level of wild-type ESCs. $n = 3$ biological replicates. **b** Colony formation assay for wild type, *Trincr1−/−* ESCs cultured in 2i + LIF or PD + LIF condition on feeder cells in a 12-well plate. Two independent *Trincr1−/−* ESC clones were analyzed. Data were normalized to the colony number of wild-type ESCs in 2i + LIF. $n = 3$ biological replicates. **c** Principal component analysis of all protein-coding genes in wild type and *Trincr1−/−* ESCs in 2i + LIF and PD + LIF. **d** KEGG pathway analysis of upregulated genes in *Trincr1−/−* ESCs in PD + LIF. Top 8 enriched pathways are shown with *P* values and fold of enrichment. **e** GSEA analysis for MAPK/ERK targets in wild type and *Trincr1−/−* ESCs in PD + LIF. The distinct peak at the end of the ranked list indicates that MAPK/ERK target genes are generally upregulated in *Trincr1−/−* ESCs. Shown are mean ± SD. For **a**, *P* values were determined by unpaired two-sided Student's *t*-test. For **b**, *P* values were determined by unpaired two-way ANOVA with two-sided Dunnett's test. For **e**, *P* value was determined by an empirical gene set-based permutation test

Trim71 expression since both mRNA and protein levels of Trim71 were not altered in *Trincr1−/−* or Trincr1 OE ESCs (Fig. 6g, h and Supplementary Fig. 6c). These data led us to hypothesize that Trincr1 may affect interaction between TRIM71 and SHCBP1. To test this, we overexpressed FLAG tagged TRIM71 and HA tagged SHCBP1 in HEK293 cell with or without Trincr1 overexpression and performed immunoprecipitation experiments with FLAG antibody. The results showed that Trincr1 overexpression significantly weakened the interaction between TRIM71 and SHCBP1 (Fig. 6i). Taken

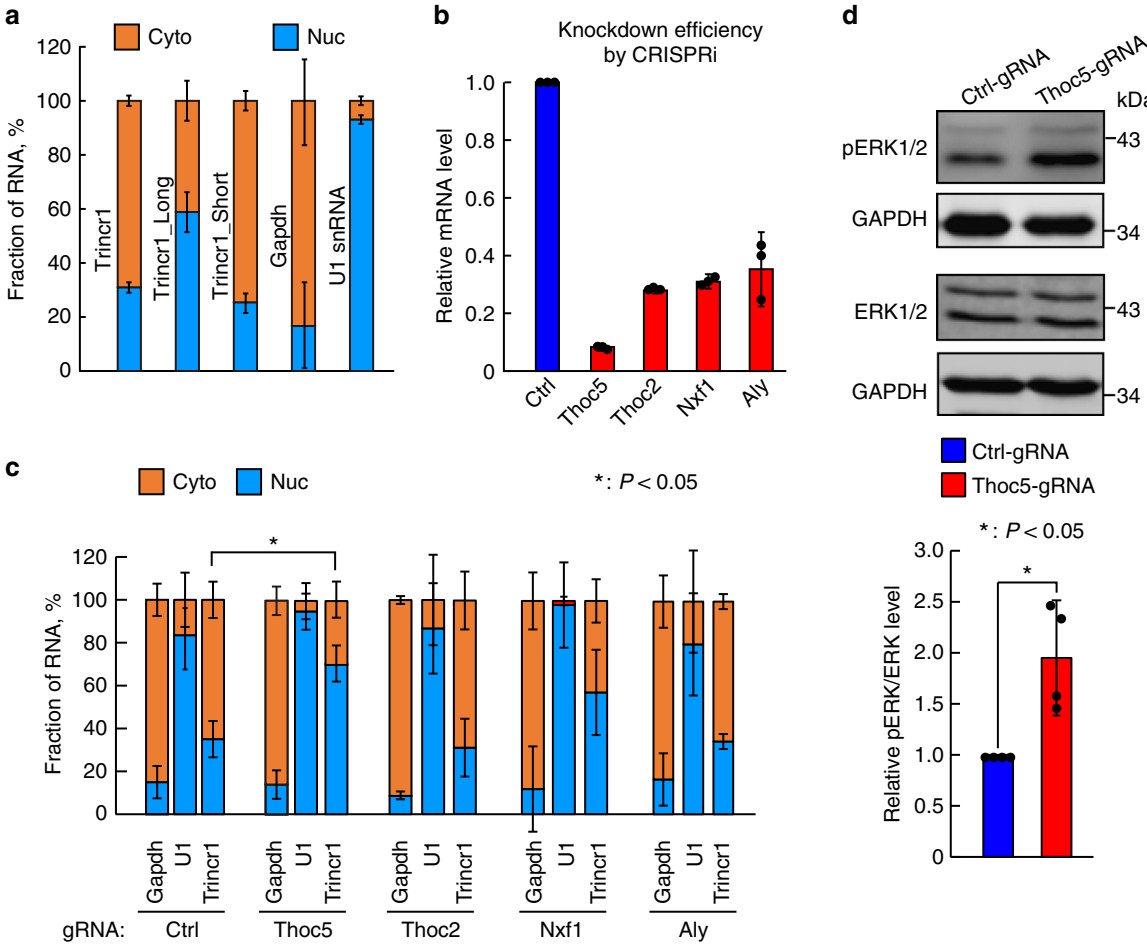

**Fig. 4** Thoc5 regulates the export of Trincr1 to cytoplasm. **a** Fraction of Trincr1 in cytoplasm (Cyto) and nucleus (Nuc). $n = 3$ independent experiments. Data were normalized by using comparable amounts of nuclear and cytoplasmic lysates from the same cell samples. U1 and Gapdh were used for the quality control of nucleus and cytoplasm fraction, respectively. **b** RT-qPCR analysis of knockdown efficiency of CRISPRi constructs targeting various components of transcription-export complex. The β-actin gene was used as a control. Data were normalized to ESCs treated with control gRNAs. $n = 3$ biological replicates. **c** Fraction of Trincr1 in cytoplasm and nucleus in CRISPRi ESCs. $n = 3$ independent experiments. Data were normalized by using comparable amounts of nuclear and cytoplasmic lysates from the same cell samples. U1 and Gapdh were used for the quality control of nucleus and cytoplasm fraction, respectively. **d** Western blotting analysis of phosphorylated ERK in control and Thoc5 CRISPRi ESCs in 2i + LIF. For quantification of pERK/ERK, data were normalized to GAPDH and then to control gRNA treated ESCs cultured in 2i + LIF. $n = 4$ independent experiments. Shown are mean ± SD. For **c**, $P$ values were determined by unpaired two-way ANOVA with two-sided Dunnett's test. For **d**, $P$ values were determined by paired two-sided Student's $t$-test

together, these data demonstrate that Trincr1 represses FGF/ERK signaling and promotes the self-renewal of ESCs through suppressing TRIM71.

**Trincr1 represses FGF/ERK signaling in NPCs**. Next, we tested whether overexpression of Trincr1 represses FGF/ERK signaling in other cells. We chose 3T3 and neural progenitor cells (NPCs) for the further test. RT-qPCR showed that Trim71 is not expressed in 3T3 cells but in NPCs (Supplementary Fig. 7a). Based on our hypothesis that Trincr1 represses FGF/ERK signaling through TRIM71, we expect that Trincr1 overexpression should only affect FGF/ERK signaling in NPCs. Indeed, overexpressing Trincr1 did not cause any change of ERK phosphorylation in 3T3 cells (Supplementary Fig. 7b, c). In contrast, overexpression of Trincr1 significantly suppressed the basal ERK phosphorylation in NPCs (Fig. 7a). Interestingly, cell morphology showed noticeable difference between control and Trincr1 overexpressing NPCs (Supplementary Fig.7d). RT-qPCR results showed that markers for progenitor cells (Nestin, Pax6) and markers for astrocyte differentiation (Aqp4, Clu, and Gfap) are

significantly downregulated and upregulated in NPCs overexpressing Trincr1 (Fig. 7b), respectively. Immunofluorescence confirmed the loss of NPC marker Nestin in Trincr1 overexpressing cells (Fig. 7c). These results are consistent with the requirement of FGF/ERK signaling in the self-renewal of NPCs. Together, these data show that ectopic expression of Trincr1 represses FGF/ERK signaling and the self-renewal of NPCs.

**Discussion**

In this study, we have identified a pair of regulators with opposing functions in fine-tuning the amplitude of FGF/ERK signaling in ESCs. TRIM71 enhances the activation of FGF/ERK signaling by stabilizing SHCBP1, while the lncRNA Trincr1 binds TRIM71 to inhibit its SHCBP1 stabilizing activity. The truncation experiments show that the physical interaction between Trincr1 and TRIM71 relies on the NHL domain of TRIM71 and the 5′ half of Trincr1. We confirmed that TRIM71 and Trincr1 work in the same axis to regulate FGF/ERK signaling by showing that knocking down Trim71 fully rescued the defects of *Trincr1−/−* ESCs. Furthermore, we found that Thoc5 is an essential

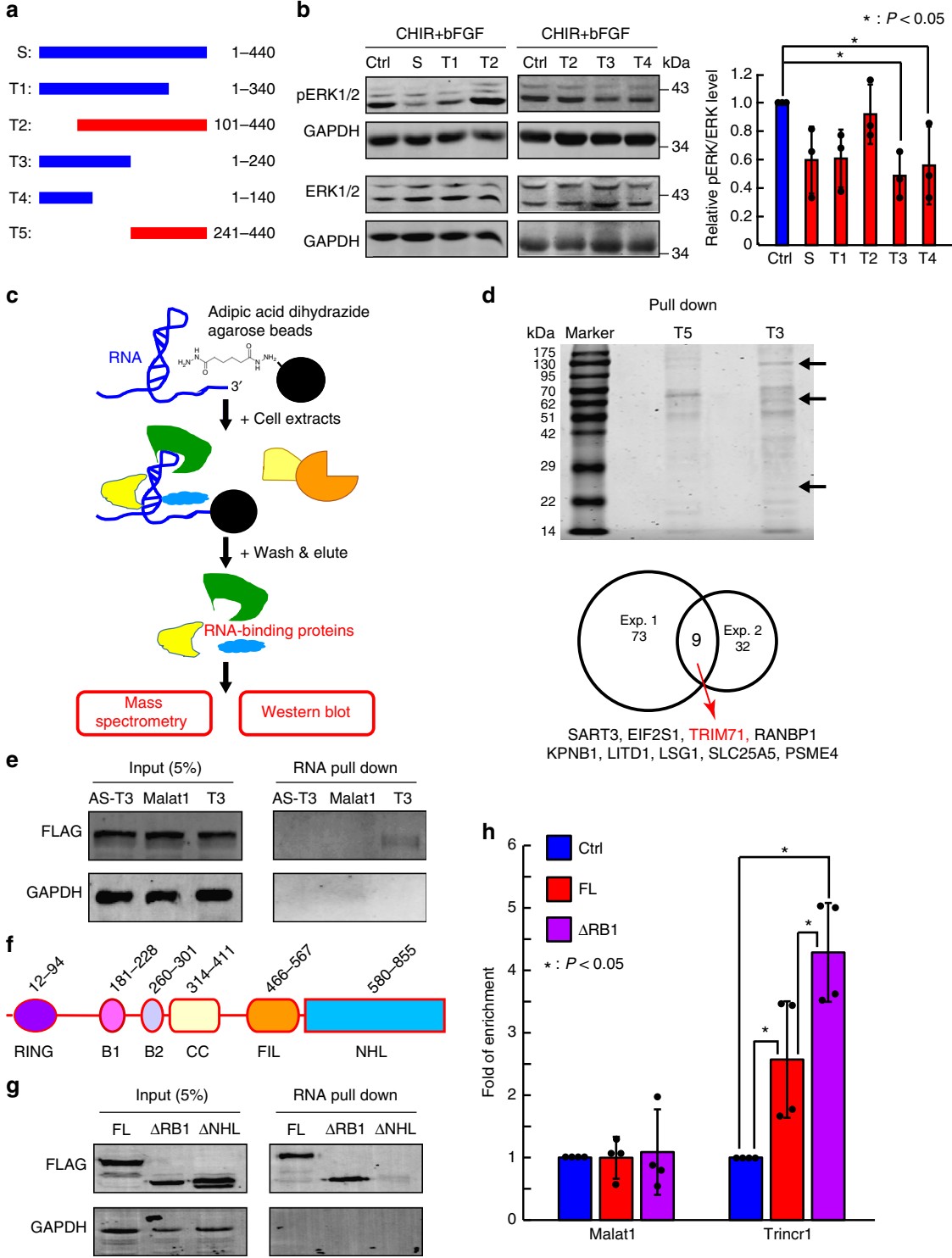

component for the nuclear export of Trincr1 and required for the suppression of ERK signaling in ESCs. More importantly, FGF/ERK signaling and self-renewal of NPCs can be manipulated by ectopic expression of Trincr1. Together, our study highlights lncRNAs as important modulators of signal transduction pathways and broadens molecular mechanisms of lncRNAs in regulating the self-renewal and pluripotency of ESCs.

Whether Trincr1 directly binds TRIM71 is currently unclear. Our data do not exclude the possibility that the interaction between Trincr1 and TRIM71 is mediated by other factors (e.g., RNA or protein). To show the direct binding between the two

molecules, biochemical experiments such as electromobility shift assay (EMSA) need to be performed with in vitro transcribed RNA and purified recombinant proteins. In addition, the identification of Trincr1 mediated regulation of TRIM71 activity suggests a broader function of lncRNAs in regulating TRIM71 related biological processes. Trim71 has been shown to be important for embryonic development, neural tube closure, reprogramming, and tumorigenesis[17–19,35,36]. In addition to its role in promoting FGF/ERK signaling, Trim71 binds and suppresses the expression of many mRNA targets including cell cycle genes[20]. More interestingly, it has recently been shown to

**Fig. 5** Trincr1 physically interacts with TRIM71. **a** Schematic design for truncated Trincr1 used in this study. Drawn in scale. Blue, RNA constructs inhibiting ERK phosphorylation; Red, RNA constructs with no effects on ERK phosphorylation. **b** Western blotting analysis of phosphorylated ERK in empty and Trincr1_S and Trincr1_S truncated overexpressing ESCs induced by bFGF. For quantification of pERK/ERK, data were normalized to GAPDH and then to control overexpression vector transfected ESCs cultured in CHIR and induced by 12 ng/ml bFGF. $n = 3$ independent experiments. **c** Experimental design for identifying Trincr1 interacting proteins. In vitro transcribed T3 was used as the bait, T5 was used as the control. **d** Trincr1 interacting proteins identified by RNA pull-down followed by mass spectrometry analysis in ESCs. Up panel shows a representative gel of RNA pull-down samples stained by Coomassie blue, and the arrow indicates the protein pulled down by T3 but not T5; Bottom panel shows the number of proteins identified by two independent pull-down experiments. **e** Western blotting analysis of FLAG tagged TRIM71 pulled down by in vitro transcribed antisense T3, a 200 nt fragment of the lincRNA Malat1, and T3 in ESCs. Shown is a representative gel of two independent experiments. **f** Graphic illustration showing different protein domains in TRIM71. **g** Western blotting analysis of Flag tagged full length (FL), RING /B1 domain truncated (ΔRB1), and NHL domain truncated (ΔNHL) TRIM71 pulled down by T3 in ESCs. Shown is a representative gel of two independent experiments. **h** RT-qPCR analysis of Trincr1 and Malat1 following Flag RIP in control, Flag-Trim71, and Flag-RING and B1 truncated Trim71 overexpressing ESCs. Data were normalized to control ESCs transfected with empty 3X FLAG overexpression vectors. $n = 4$ independent experiments. Shown are mean ± SD. For **b**, $P$ values were determined by unpaired two-way ANOVA with one-sided Dunnett's test. For **h**, $P$ values were determined by unpaired two-way ANOVA with Tukey's test

promote the function of miR-290/302 clusters by interacting with AGO2[23]. To what extent these other Trim71 regulated processes are controlled by lncRNAs warrants further studies in the future. More importantly, future work should unveil the functional impact of the interaction between lncRNAs and TRIM71 in other biological contexts, including neural development, cancer, and reprogramming.

Our study highlights the concept of physical interactions between signaling regulators and lncRNAs for fine-tuning the amplitude of signaling to coordinate cell fate specification. In this case, Trincr1 suppresses the function of its protein partner Trim71. However, whether RNA can promote the function of Trim71 is not known. How general the regulation of signaling pathway by RNA remains unclear so far. As shown by us in NPCs, lncRNAs may provide efficient means to control cellular signaling and cell fate. Importantly, RNA interactome studies have identified many signaling proteins as non-canonical RNA binding proteins in various cell types[37–39], which may serve as platforms for lncRNAs to regulate the cognate signaling pathways. Future work should unveil the hidden RNA world in signaling pathways and its function in regulating cell fate in development and disease. These lncRNAs may provide valuable tools and targets for regenerative medicine and cancer therapy.

## Methods
**Cell culture**. Mouse ESCs were cultured on 0.1% gelatin-coated plates in 2i + LIF medium, which consist of N2B27 supplemented with MEK inhibitor PD0325901 (1.0 µM), GSK3 inhibitor CHIR99021 (3.0 µM), and leukemia inhibitory factor (1,000 unit per ml). For PD + LIF experiments, ESCs were first plated in 2i + LIF for 24 h, then washed once with 1xPBS and changed to N2B27 supplemented with PD + LIF, cultured for another four days before analysis with medium changed every other day; For colony formation assay, 600 cells were plated on a 6-well 0.1% gelatin-coated plate in 2i + LIF medium, or 300 cells were plated on feeder cells in a 12-well plate in 2i + LIF medium; Colonies were counted 5–7 days after plating under a dissection microscope. For the derivation of NPCs, ESCs were dissociated and cultured in low-adherent plates to form embryoid bodies (EB). EB kept in suspension culture for 4 days were then plated onto poly-ornithine and laminin-coated plate. Twenty four hours after plating, medium was switched to ITSFn medium and replenished every 2 days. After 6–8 days, cells were re-plated into N2 medium supplemented with 20 ng per ml bFGF on dishes pre-coated with poly-ornithine and laminin. Following 1–2 passages, 20 ng per ml EGF together with FGF were added into N2 medium to stimulate neural progenitor specification. All cell lines were tested to avoid mycoplasma contamination regularly in the lab. Wild-type mouse ESCs were previously generated by us[13]. The HEK293 cells were from ATCC.

**RNA extraction and RT-qPCR**. Total RNA was extracted following standard TRIzol protocol (Invitrogen). RT-PCR were performed using SYBR Green mix (Vazyme Biotech, Nanjing). Sequences for RT-qPCR primers are shown in Supplementary Table 1. To quantify the ratio of Trincr1_L versus Trincr1_S, both the common (L plus S) and long isoform (Trincr1_L) amplicons were cloned in tandem into the same plasmid. Five different dilutions (1 fg per µl to 10 pg per µl) were made for qPCR to make the standard curve. The amount of Trincr1_L and total Trincr1 was calculated by fitting the Ct value into the respective standard

curve. The amount of Trincr1_S was calculated by subtracting the value of Trincr1_L from the value of total Trincr1.

**siRNAs, shRNA, and CRISPRi design**. siRNAs (GenePharma, Shanghai) were transfected at 50 nM using the DharmaFECT1 transfection reagent (Dharmacon, GE Healthcare) following the manufacturer's protocol. pLKO or pSicoR vectors were used to make lentiviral shRNA constructs. For the CRISPRi experiment, we designed sgRNAs based on the website http://crispr-era.stanford.edu. The control CRISPRi vector had an 18 bp non-targeting guide: 5′-GGGTCTTCGAGAA-GACCT-3′. The U6 promoter driven sgRNA sequences were cloned into a piggybac vector expressing KRAB-dCas9 with Hygromycin B resistance. After transfection, CRISPRi ESCs were obtained after 2 weeks selection under 130 µg per ml Hygromycin B and characterized by RT-qPCR. Sequences for siRNAs, shRNAs, and CRISPRi sgRNAs are listed in Supplementary Table 2.

**Trincr1 knockout and overexpression**. To knockout Trincr1, a pair of guide RNA sequences were designed by the tool from the website http://crispr.mit.edu/. The Guide RNA sequences were: g1, 5′-GACAUGCUUGGUAACUAUGU-3′; g2, 5′-GAAGCAGAACCCAGACCAAC-3′. In order to overexpress Trincr1, Trincr1_L, and Trincr1_S was amplified from cDNA of ESCs and cloned downstream of CAGGS promoter in piggybac transposon expression vectors digested with BamhI and NheI. Approximately 1 µg piggybac Trincr1_L or Trincr1_S plasmid with 1 µg pBase plasmid were transfected to V6.5 ESCs. After 5 days selection under 130 µg per ml hygromycin B, 300 cells were plated to one well of 6-well plate for colony picking.

**Western blot analysis**. Antibodies against ERK (#9102), pERK (#9101) were from Cell Signaling Technology, against GAPDH (MB001) was from Bioworld Technology (Nanjing, China). Against FLAG (F1804) was from Sigma. Anti-rabbit (926–32213) and mouse (926–68022) secondary antibodies were from LI-COR Biosciences and membranes were imaged using Oddssey. For background pERK analysis in 2i + LIF and PD + LIF condition (Figs. 2d, 2f and 6b), because the signal was too weak, HRP-conjugated anti-rabbit secondary antibodies were used and membranes were incubated with the Western ECL Substrate (WBKLS0500, Milipore) and imaged using Amersham imager 600 (GE Life Sciences). All primary antibodies were used at a dilution of 1:1000. All secondary antibodies were used at a dilution of 1:10000. For the analysis of pERK induced by bFGF, cells were plated in 2i + LIF for ~24 h, then grown in CHIR (to remove the interfering effects of PD and LIF on ERK phosphorylation) for ~48 h before bFGF induction, proteins were extracted 15 min after the addition of 12 ng per ml bFGF. Unprocessed images for all western blots in main figures are shown in Supplementary Figure 8.

**Immunofluorescence staining**. For immunofluorescence, cells were fixed in 4% PFA at room temperature for 15 min. Then cells were permeabilized with PBS containing 0.2% Triton-X100 and blocked in PBS containing 2% BSA. Incubation for Nestin primary antibody (NES, Aves Labs) was done in blocking solution at 4 °C overnight. Then incubate cells with secondary antibody (ab150169, Abcam) for 1 h at room temperature in the dark. Both primary and secondary antibodies were used at a dilution of 1:500, and cells were counterstained with DAPI.

**CRISPR mediated FLAG knockin at endogenous *Trim71* locus**. We were not able to obtain a good quality TRIM71 antibody. To detect endogenous TRIM71 protein level in Fig. 6h, we knocked in a 3XFLAG sequence before the stop codon of Trim71. The targeting strategy is illustrated in Supplementary Figure 6c. The sgRNA was designed by the website http://crispr.mit.edu/ and the sequence is 5′ CATCTTCTAATTGTGTCTTC-3′. The U6 promoter driven sgRNA sequence was cloned into a piggybac vector expressing Cas9 with hygromycin B resistance, and 5′ (778 bases) and 3′ (694 bases) arms were cloned in to a targeting vector flanking 3X

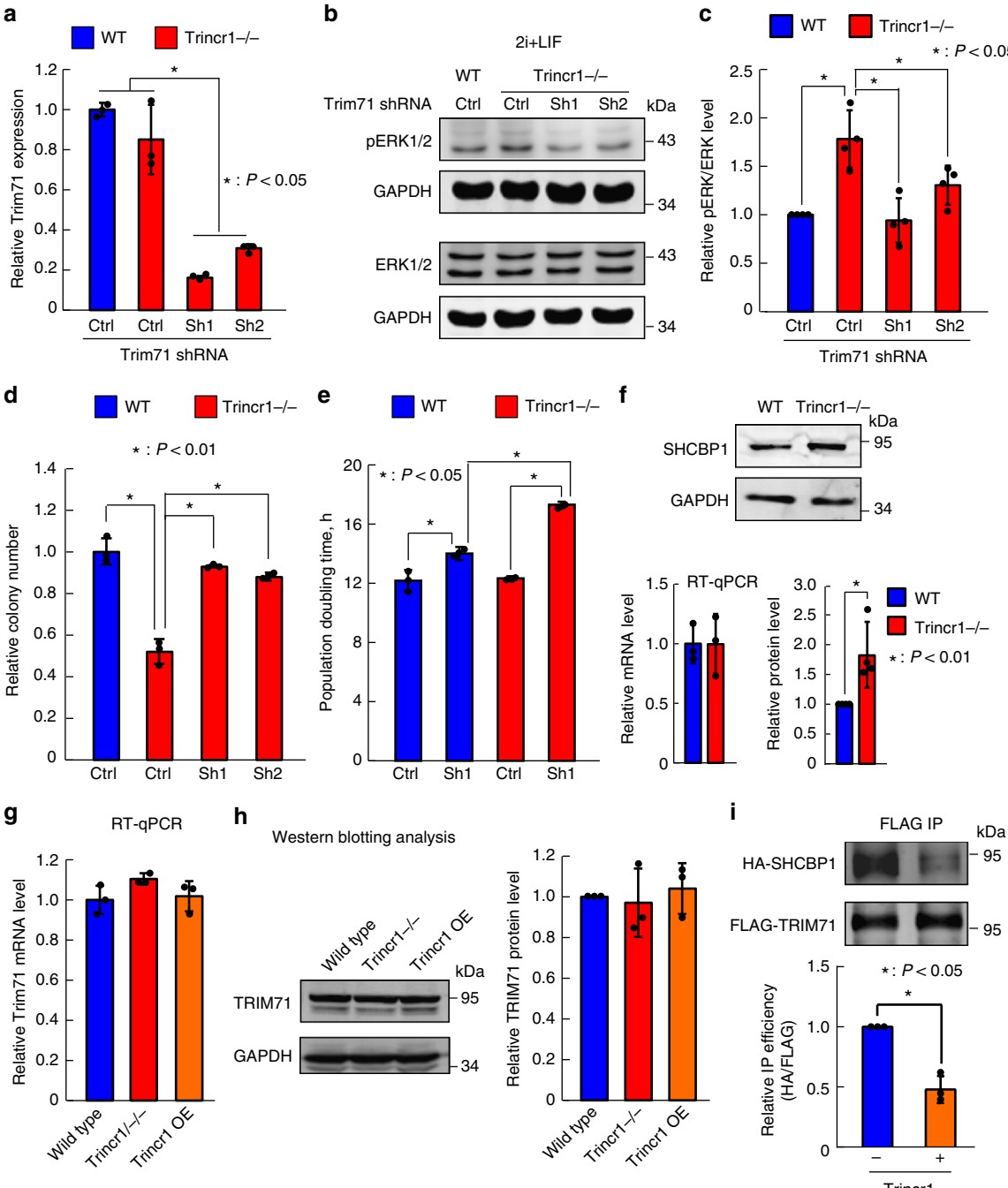

**Fig. 6** Trincr1 functions through suppressing TRIM71. **a** RT-qPCR analysis of Trim71 knocking down in *Trincr1−/−* ESCs. $n = 3$ biological replicates.
**b** Western blotting analysis of pERK in control shRNA vector treated wild-type ESCs and control or Trim71 shRNA vector treated *Trincr1−/−* ESCs cultured in 2i + LIF. Shown are representative images. **c** Quantification of pERK/ERK. $n = 4$ independent experiments. **d** Colony formation assay for control or Trim71 shRNA ESCs in wild type or *Trincr1−/−* in PD + LIF. $n = 3$ biological replicates. **e** Population doubling time for *Trincr1−/−* and Trim71 knockdown ESCs in 2i + LIF medium. $n = 3$ biological replicates. The first two columns were the same data appeared in Supplementary figure 1e. **f** Western blotting analysis and qPCR of Shcbp1 in wild type and *Trincr1−/−* ESCs. Bottom left: quantification of western blot data, $n = 4$ independent experiments. Bottom right: qPCR of Shcbp1. Data were normalized to wild-type ESCs. $n = 3$ biological replicates. **g** RT-qPCR analysis of Trim71 expression in wild type, *Trincr1−/−*, and Trincr1_S overexpressing ESCs. Data were normalized to wild-type ESCs. $n = 3$ biological replicates. **h** Western blotting analysis of FLAG knockin TRIM71 in wild type, *Trincr1−/−*, and Trincr1_S overexpressing ESCs in 2i + LIF. $n = 3$ independent experiments. **i** Western blotting analysis of immunoprecipitation with FLAG antibody (FLAG IP) in HEK293 cells expressing HA-SHCBP1 and FLAG-TRIM71 with or without Trincr1_S overexpression. Shown are representative images of IP samples (left) and quantification of IP efficiency (right). IP efficiency was calculated as HA-SHCBP1 /FLAG-TRIM71 and then normalized to samples without Trincr1_S overexpression. $n = 3$ independent experiments. For the RT-qPCR of **a**, **f**, and **g**, the β-actin gene was used as a control. Shown are mean ± SD. For **a**, **c**, and **d**, data were normalized to wild-type ESCs treated with control shRNA vectors, *P* values were determined by unpaired one-way ANOVA with two-sided Dunnett's test. For **e**, *P*-values were determined by unpaired two-way ANOVA with Tukey's test. For **f** and **i**, *P* values were determined by paired two-sided Student's *t*-test

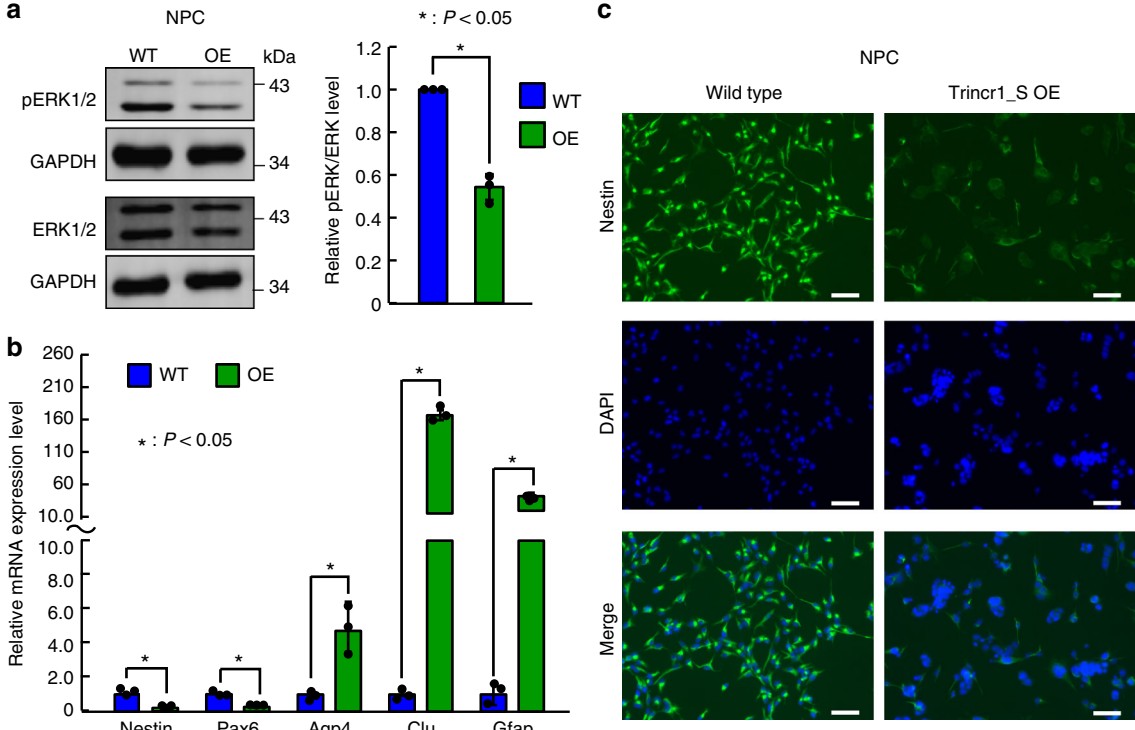

**Fig. 7** Ectopic expression of Trincr1 represses FGF/ERK signaling in NPCs. **a** Western blotting analysis of pERK in NPCs overexpressing Trincr1_S. Shown are representative images (left) and quantification (right). Data were normalized to ERK and then to wild-type NPCs. $n = 3$ independent experiments. **b** RT-qPCR analysis of markers for NPCs and astrocytes. The β-actin gene was used as a control. Data were normalized to the mRNA level of wild-type NPCs. $n = 3$ biological replicates. **c** Immunofluorescence of Nestin in wild type and Trincr1_S overexpressing NPCs. Green represents Nestin, Blue represents DAPI. Scale bars, 100 μm. Shown are mean ± SD. For **a**, P values were determined by paired two-sided Student's t-test. For **b**, P-values were determined by unpaired two-sided Student's t-test

FLAG T2A GFP sequence. After transfection, CRISPR knockin ESCs were obtained after selection for 2 weeks under 1 μg per ml puromycin and characterized by genomic PCR and sequencing. The primers for genomic PCR locate outside of 5′ arm and 3′arm as shown in Supplementary Figure 6c. The expected PCR products are 2475 bp for knockin allele and 1737 bp for wild-type allele.

**Subcellular fractionation**. The subcellular fractionation assay was performed as a previously published procedure with modifications[40]. Cells grown in 10 cm dishes were digested with 2 ml Accutase (Invitrogen) for 5 min at 37 °C and were then pelleted at 1000 rpm for 5 min and re-suspended in 4 ml of ice-cold 1 × PBS. The resuspension was centrifuged at 1000 rpm for 5 min at 4 °C. The supernatant was carefully removed without disturbing the pellet. The pellet was re-suspended in five packed pellet volumes of ice-cold cytoplasmic extraction buffer (20 mM Tris, pH 7.6, 0.1 mM EDTA, 2 mM $MgCl_2$). The cells were incubated first at room temperature for 2 min, then on ice for 10 min. The cells were lysed by addition of CHAPS to a final concentration of 0.6%. The sample was then homogenized by passage through a 1 ml syringe about 40 times and centrifuged at 500 × g for 5 min at 4 °C. Around 70% of the supernatant was taken and saved at −80 °C; This was the cytoplasmic fraction. The remaining supernatant was carefully removed, and the pellet was washed with cytoplasmic extraction buffer supplemented with 0.6% (w/v) CHAPS. The sample was centrifuged at 500 × g for 5 min at 4 °C, and the entire supernatant was discarded. The wash step was repeated one more time. The pellet was then re-suspended in two packed pellet volumes of nuclei suspension buffer (10 mM Tris, pH 7.5, 150 mM NaCl, 0.15% (v/v) NP–40). The nuclear suspension was layered on five packed pellet volumes of sucrose cushion (10 mM Tris, pH 7.5, 150 mM NaCl, 24% (w/v) sucrose) and pelleted at 14,000 rpm for 10 min at 4 °C. The supernatant was discarded, and the pellet was washed with ten packed pellet volumes of ice-cold 1 × PBS supplemented with 1 mM EDTA. The sample was then centrifuged at 500 × g for 5 min at 4 °C. The pellet constituted the nuclear fraction. For RT-qPCR to determine the cytoplasmic/nuclear distribution of Trincr1, normalization was done using comparable fraction of cytoplasmic and nuclear lysates from the same cell samples.

**UV crosslink RIP**. The cells grown in a 10 cm dish were crosslinked with 254 nm UV (400 mJ per cm2). Cells was collected and lysed by lysis buffer (50 mM Tris HCl, pH 7.4, with 150 mM NaCl, and 1% TRITON X-100, 5% glycerol, supplemented with 1 mM DTT, 1 mM PMSF, 1:500 PI cocktail, and 400 U per ml RNase

Inhibitor). The lysate was treated with DNase I at 37 °C for 10 min, then was centrifuged at 12,000 rpm for 20 min at 4 °C. The supernatant was incubated with 20 μl protein A/G Dynabeads (Invitrogen) at 4 °C for 1 h to avoid non-specific binding proteins. Meanwhile 5 μg Flag antibody was incubated with 30 μl protein G Dynabeads (Invitrogen, 10003) at room temperature for 30 min in the dilution buffer (50 mM Tris-Cl, pH 7.4, 150 mM NaCl, 1 mM EDTA, 0.1% TRITON X-100); 100 ng yeast total extract were then added to block beads for at least 1 h before use. Pre-cleared lysate was added to the clean Flag-beads complex, and mixed gently overnight at 4 °C on a shaker. After incubation, the samples were washed five times with 0.5 ml of IP200 buffer (20 mM Tris-Cl pH7.4, 200 mM NaCl, 1 mM EDTA, 0.3% TritonX-100, 5% glycerol), followed by digestion with Proteinase K for 1 h. RNA were then extracted with TRIzol as described above.

**RNA pull-down and mass spectrometry analysis**. The RNA pull-down assay was modified from a published procedure[41]. Substrate RNA for beads immobilization were synthesized by in vitro transcription using T7 RNA polymerase (Thermo, EP0111) and PCR products as templates. A volume of 250 pmol RNA was used in 400 μl reaction mixture containing 100 mM sodium acetate (pH 5.0) and 5 mM sodium meta-periodate (Sigma, 71859). Reaction mixtures were incubated 1 h at room temperature avoiding light. The RNA was then ethanol precipitated and re-suspended in 500 μl 0.1 M sodium acetate (pH 5.0). About 200-μl adipic acid dihydrazide agarose beads (Sigma, A0802) were washed four times with 0.1 M sodium acetate (pH 5.0), pelleted after each wash at 1000 rpm for 3 min. After the final wash, the beads was re-suspended in 300 μl 0.1 M sodium acetate (pH 5.0) and mixed with the periodate-treated RNA, rotated overnight at 4 °C. The beads with the bound RNA were then pelleted and washed three times in 1 ml of 2 M NaCl and three times in 1 ml of NP40 buffer (50 mM Tris, pH 8.0, 150 mM sodium chloride, 1.0% NP-40, 1 mM EDTA supplemented with 1 mM DTT, 1 mM PMSF, 1:500 PI cocktail, and 400 U per ml RNase Inhibitor).The beads containing immobilized RNA were then incubated with cell lysates 20 min at 30 °C. After incubation, the beads were pelleted by centrifugation at 1000 rpm for 3 min and then washed twice with NP40 buffer and twice with high salt NP40 buffer (350 mM NaCl). The protein bound to the target RNA were eluted by heating the beads at 95 °C for 5 min with 30 μl SDS loading buffer. For LC-MS/MS analysis, the eluted peptides were sprayed into a Velos Pro Orbitrap Elite mass spectrometer (Thermo Scientific, USA) equipped with a nano-ESI source. The mass spectrometer was operated in data-dependent mode with a full MS scan in FT mode at a resolution of

120,000 followed by CID (Collision Induced Dissociation) MS/MS scans on the 15 most abundant ions in the initial MS scan.

**RNA-Seq and bioinformatics**. Total RNA was subjected to two rounds of purification using poly-T oligo-attached magnetic beads before the synthesis of double-stranded (ds) cDNA. RNA from RIP experiment was directly used for the synthesis of ds-cDNA. The ds-cDNA was ligated to adaptors and sequenced using Illumina Genome Analyzer (Novogene). The reads were aligned to the mouse genome (mm9) with STAR (version 2.5.0) using the GENCODE transcript annotation as transcriptome guide. All programs were used with default settings unless otherwise specified. Expression levels were quantified as normalized FPKM using Cufflinks. For RIP-seq, reads aligned to non-polyA transcripts were excluded. To calculate fold enrichment for RIP, we normalized the FPKM value of every transcript to the average of all transcripts. GSEA2.4 was used to test for the enrichment of selected gene sets by java GSEA Desktop Application. R 3.1.1 was used for the generation of scatter plot and boxplot. Gene ontology and KEGG pathway analyses were performed using DAVID.

**Co-IP of SHCBP1 and TRIM71**. For SHCBP1 and TRIM71 Co-IP, 5 million HEK293 cells were plated to a 10 cm dish. A volume of 4 µg 3X FLAG-TRIM71, 6 µg 3X HA-SHCBP1, and 12 µg Trincr1 expression vectors was transfected ~20 h after plating. Forty eight hours after transfection, the cells were harvested and lysed in Co-IP lysis buffer (50 mM Tris-HCl, 140 mM NaCl, 0.5% NP-40, 1 mM EDTA, 10% Glycerol supplemented with 1 mM DTT, 1 mM PMSF, 1:500 PI cocktail). After centrifugation, 1% supernatant were collected as input, the remaining was incubated with 5 µl FLAG antibody and 30 µl protein G Dynabeads for 12 h at 4 °C. After washing five times with lysis buffer, the beads were collected for western blot analysis.

**Statistical analysis**. The data were presented as mean ± SD except where indicated otherwise. We performed two-tailed Student's t-test to determine statistical significance. For multiple comparison, we performed one-way or two-way ANOVA followed by Dunnett's test or Tukey's test as indicated in figure legends. For analysis shown in the boxplot graph, we performed two-tailed Wilcoxon signed-rank test. P-value < 0.05 was considered as statistically significant.

**Reporting Summary**. Further information on experimental design is available in the Nature Research Reporting Summary linked to this article.

## Data availability

All data generated or analyzed during this study are included in the manuscript and its supplementary information files. RNA-seq and RIP-seq data are deposited in NCBI's Gene Expression Omnibus under the accession GSE125458. Proteomics data can be accessed on PRIDE (PRoteomics IDEntifications) database under the accession number: PXD012493. All data that support the findings of this study are available from the corresponding authors upon request.

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

## Acknowledgements

We would like to thank members of Wang laboratory, Drs. Ge Shan and Yang Yu for critical reading and discussion of the manuscript. We thank Dr. Richard Gregory for sharing full-length Trim71 cDNA and the mass spectrometry facility of National Center for Protein Sciences at Peking University for assistance with LC-MS/MS analysis. We thank Dr. Xiaoqun Wang for sharing Nestin antibody. This study was supported by The National Key Research and Development Program of China [2016YFA0100701 and 2018YFA0107601] and the National Natural Science Foundation of China [91640116, 31471222, 31622033 and 31821091] to YW.

## Author contributions

Y.-P.L. and F.-F.D. performed all experiments with help from other authors. Y.-T.Z. performed experiments in Fig. 4 with help from F.-F.D. K.-L.G. and J.H. performed bioinformatics analysis. L.-Q.L. and H.-B.S. helped construct overexpression plasmids. K.Z. and N.Y. provided assistance for TRIM71 and Trincr1 interaction experiments. All authors were involved in the interpretation of data. Y.W. conceived and supervised the project and wrote the manuscript with help from Y.-P.L., F.-F.D., Y.-T.Z., and K.-L.G.

## Additional information

**Competing interests:** The authors declare no competing interests.

