## [Peer Review File · Nature Communications]

Reviewers' comments:

Reviewer #1 (Remarks to the Author):

In this manuscript, the authors identified a TRIM71-interacting long noncoding RNA, which they named *Trincr1*, that plays an important role in FGF/ERK signaling and progenitor cell self-renewal. They showed that *Trincr1* is bound by *Trim71* in mouse embryonic stem cells (ESCs), via its 5'-region and the NHL domain in *Trim71*. It negatively regulates ERK signaling, and is required for ESC identity and supports ESC self-renewal in sub-optimal culture conditions. *Trincr1* likely functions in the same pathway with *Trim71*, as *Trim71* depletion rescues the defects of *Trincr1* knockout ESCs. Finally, they showed that *Trincr1* also represses FGF/ERK signaling in neural stem cells (NSCs) and is a negative regulator of NSC maintenance. Together, this study uncovered a novel regulator *Trincr1* in FGF/ERK signaling and highlights lncRNAs as important players in stem cell fate decisions.

While the role of *Trim71* and *Shcbp1* in ERK signaling and ESC maintenance has been previously reported, the identification of *Trincr1* as well as its functional interaction with *Trim71* are novel findings. To further strengthen the manuscript, the authors are encouraged to address the following questions and comments:

1. Based on the proposed model, *Trincr1* and *Trim71* play important roles in FGF/ERK signaling. However, many of the experiments in the manuscript were carried out in culture conditions in which ERK signaling was blocked by PD0325901. The addition of the inhibitor may mask or undermine some of the observations, and it is therefore important to repeat some of the key experiments in culture conditions where FGF/ERK signaling is not inhibited (such as serum+LIF).
2. To provide further support for the proposed mechanism, it will be important to test the effect of *Trim71* overexpression in *Trincr1* $-/-$ ESCs, the effect of *Trincr1* KO on *Trim71* expression, and the effect of *Trincr1* KO on the interaction between TRIM71 and SHCBP1.
3. *Trim71* has been shown to promote ESC proliferation (PMID: 22735451). Based on this study, *Trincr1* represses *Trim71*. However, *Trincr1* also promotes ESC maintenance. Can the authors provide some explanations to reconcile these findings?

Minor points:

Fig 1A, 1B: Consider moving them to the supplemental figures, as these observations do not promote the main conclusions in the paper and they have been reported previously in neural progenitor cells (PMID 22508726).

Fig 1D, 1E: Can the authors provide browser shots and RIP-qPCR data for additional *Trim71*-bound RNAs? There is a clear difference in fold-enrichment for *Trincr1* between RIP-seq and RIP-qPCR (~70 vs ~6).

Fig 2G: How much fold-increase was *Trincr1* overexpression compared to the endogenous level?

Fig 3C, 3D: Consider removing 3C, as it presents the same information as 3D.

Fig 5H: FL-*Trincr1* RIP-seq signal appears to be much lower than what was in Fig 1E. Interestingly, delta-RB mutant appears to show stronger interaction with *Trincr1*. Can the authors comment on these observations?

Fig 6B, 6C: There is some visual inconsistency in pERK/ERK between the blot in 6B and the quantitation in 6C for *Trim71*-shRNA2: sh2 appears to reduce pERK level more substantially than sh1 in 6B, but that is not the case in 6C.

Fig 6: Will disruption of the NHL-domain in *Trim71* also rescue the *Trincr1* deletion phenotype?

Fig 7: The authors referred to the ESC derived neural stem cells as GPCs. However, they are really NSCs based on a previous report (PMID: 16086633) and the fact that they express neural progenitor markers *Nestin* and *Pax6*. Need to clarify the nomenclature.

Fig 7A: "The result is as expected since these cells does not express any *Trim71*". Can the author compare the expression of *Trim71* between 3T3 and ESCs?

Fig 8: Consider changing ESC to Progenitor Cells, so that the model can encompass results from both

ESCs and NSCs.

Other wording changes:

1. "Together, these data show that similar to its role in neural progenitor cells, Trim71 promotes FGF/ERK signaling through upregulating SHCBP1 protein in ESCs". The causality has not been fully established at this point.
2. "qRT-PCR analysis for a panel of pluripotency markers and early differentiation marker Fgf5 confirmed that Trincr1 is important for the self-renewal and pluripotency of ESCs in PD+LIF". Marker analysis only shows the requirement for Trincr1 in the maintenance of ESC identity, not necessarily in self-renewal and pluripotency.
3. Similarly, "Together, these data demonstrate that Trincr1 is important for the self-renewal and pluripotency of ESCs cultured in various sub-optimal conditions". The data showed that Trincr1 is important for self-renewal, but did not provide sufficient information on its role in pluripotency.

Reviewer #2 (Remarks to the Author):

Li et al present an interesting new study describing a novel role for the lncRNA TRINCR in fine-tuning cellular responses to FGF signaling in ESCs. They use a wide variety of appropriate molecular approaches to identify a putative (though not necessarily direct) interaction between TRINCR and TRIM71 – a known modulator of FGFR activity. The study builds in a logical fashion, starting with the identification of a modest effect of TRINCR knockdown on ERK signaling, then using RIP-seq to identify TRIM71 as a potential binding partner.

The introduction is rather brief - no discussion of TRIM71 – this could be expanded upon, rather than just coming up in the discussion.

One weakness of the manuscript is the failure to display direct binding between TRINCR and TRIM71 – they need to mix recombinant protein with IVT RNA and show that the two interact.

Taking each of the figures in turn

Fig 1

Why do they use FGF2 and not the more relevant FGF4?

The argument for quantitating ERK and P-ERK on different blots is very odd – not one I've ever come across in signaling papers and when the differences are so minor it would be nice to see data from the same blot. They also chop and change between quantitating with numbers and showing graphs of quantitation. In (b) there is no mention of the vast difference between lanes 1+4 compared to (a) lanes 1+3; these should be very similar as they are the exact same conditions. No quantitation in (b) and n=2 – this is a common problem throughout – sometimes n=2, sometimes 3.

Fig 2

(d) blots are Chir+FGF2 not 2i+LIF. The differences between Sh1+2 show why it is important to look at more than one (unlike in Sup Fig 1(a))

Fig 3

Why does SLE appear in both up and down regulated in (e)?

Fig 4

(c) needs controls for fractionation efficiency. Nfx1 must be repeated to get n=3

Fig 5

(b) needs a graph of quantitation and data for T5. F+h) colour schemes are very confusing – unnecessary. Need to comment on why deltaRB gives more TRINCR1 enrichment compared to FL. D) T5 not T6 in legend.

Fig 6

a) Explain why Trincr1^{-/-} has less Trim71 than WT.

Fig 7

They need to present RTPCR evidence to back up their claim that 3T3 cells don't express Trim71 and GPCs do. Could do with more informative labeling in (a). c) is very uninformative – Immunofluorescence for a differentiation marker would be better

Fig 8

Not very attractive figure. They don't show any evidence of action on FGFR2 directly throughout the paper and no evidence of direct interaction between Trincr1 and TRIM71. No mention of Ub elsewhere – need to test this with proteasome inhibitors.

Sup Fig 1

Need to test more than one ShRNA. D) cannot be true – how to go from 200,000 to 5 million cells in 2 days. They have only measured one timepoint so why the line graph?

Throughout paper all blots need kDa

Sup Fig 3

Why only n=2? What is purple dotted line in (a)? Scale bars in (b). Error bars in (d) . Stain in d is what?

Sup Fig 4

How is the MEK pathway 1.8x upregulated – not according to the figure?

Reviewer #3 (Remarks to the Author):

In this manuscript, the authors identified a lincRNA that binds to TRIM71 and has a potential role in maintaining ESCs pluripotency in the context of FGF/ERK signalling.

It has been reported previously that TRIM71 promotes FGF/ERK signalling by stabilizing SHCBP1. In the manuscript, the authors showed that the knockdown of Trim71 by shRNA and siRNA results in a reduction of p-ERK and the expression of SHCBP1. They also found that TRIM71 binds at least three lincRNAs and explored in more detail the interaction with Trincr1. They show that TRIM71 NHL domain interacts with the 1-140nt of Trincr1 and that it is exported into the cytoplasm by ThOC complex. Overall, the authors present some reasonable evidence to suggest a role for Trincr1 in controlling ERK activity levels in ES cells. However, as a lot of the effects are seen when ERK signalling is blocked through MEK inhibition, it is possible that Trincr1 works via a different route. Hence there are a number of areas that require further attention.

Major issues:

(1) Throughout the paper, the effects on ERK phosphorylation are difficult to make out. The authors do provide quantification but this is from a single repeat. As the experiments have presumably been repeated multiple times, the authors should provide averages of these experiments (with associated

error ranges).

(2) In Figure 1, the authors show a reduction on Trim71 mRNA and these leads to a reduction in the levels of SHCBP1. The knock down of Trim71 and Shcbp1 result in a reduction of the levels of p-ERK. However, the authors do not address what effect does this has on pluripotency and self-renewal. I think this is important for assessing the effect of Trincr1 in this pathway.

(3) In Fig. 1b, virtual complete loss of SHCBP1 has a similar effect on P-ERK levels compared to <50% depletion in Fig. 1a. Why is this?

(4) Did the authors run controls for Fig. 1c? ie were replicates run? Was an IP from the parental cell line done? Some validation of Trincr1 is done but none of the others are validated. Therefore if no replicates are done etc, then it should be made clear that all the other bound RNAs should be considered preliminary and Fig. S1f is essentially meaningless.

(5) In Fig. 1e, rather than using n=2, of presumably technical replicates, the authors should provide the average of the 3 independent experiments they refer to. Note that statistics cannot be done on n=2.

(6) The authors show that Trincr1 binds TRIM71 and that differentiating cells show a reduction of Trincr1 expression (others have also shown a reduction in TRIM71 expression upon differentiation). However, the levels of TRIM71 during differentiation are not assessed. I think that if Trincr1 is having an effect on differentiation through TRIM71 it would be important to show the levels of the protein also vary. (In previous reports it has been shown that the levels of TRIM71 are reduced upon differentiation of Neural progenitor cells).

(7) In Fig. 2f, the authors show that P-ERK is detectable in 2i+Lif conditions. However this should not be possible due to the MEK inhibitor being present. Indeed this is what the authors themselves show (see Fig. 1a). Please explain. The same applies to Fig. 1d. Also, how can increases in P-ERK levels be seen in the presence of PD inhibitor, as there should be no activation from MEK present. Similarly in Fig. 3a, Trincr1 has an effect but MEK (and hence ERK) will be inhibited under these conditions. This suggests that Trincr1 might be working through an alternative route to the ERK signalling pathway.

(8) If always working in the same way, why do different genes respond to Trincr1 depletion under different conditions in Supplementary Fig. 3a?

(9) In figure 4, they show that Thoc5 is responsible for exporting Trincr1 into the cytoplasm. They also show that the Thoc5 knockdown has an effect on phosphorylation of ERK but the control that they use is only mock transfected cells. It would be more appropriate to use a non-targeting guide or even one of the other guides that had no effect on the levels of cytoplasmic Trincr1, in this way they can rule out any effect of the CRISPRi on p-ERK.

(10) Fig. 4 is peripheral to the main story and could be moved to the Supplementary data.

(11) The truncation experiments in figure 5b lack the levels of p-ERK in the T5 fragment, which should be similar to the ones of T2. I think it is important to show this as they use this fragment as control in further experiments.

(12) In figure 6a, the authors show that the levels of TRIM71 mRNA in Trincr1 -/- cells are reduced compared to wild-type cells. They also show an increase in the levels of SHCBP1 protein (but not mRNA) in the knockout cells (Fig. 6f). This is contrary to Figure 1 where they show that the levels of SHCBP1 go down when TRIM71 is downregulated. If Trincr1 is a negative regulator of TRIM71, the levels of the proteins would need to be assessed to see that the effect they are observing is not independent form TRIM71.

(13) In figure 3a, they show that the loss of Trincr1 diminished the expression of pluripotency genes. In contrast, RT-qPCR results in figure 7c show that markers for progenitor cells are significantly downregulated in GPCs overexpressing Trincr1. Is Trincr1 having opposite functions in ESCs and GPCs?

(14) What effect does Trincr1 has on TRIM71? And how do they know that the effects they see are because of the binding of Trincr1 to TRIM71?

Minor comments:

To definitively state that trincr1 is not a coding RNA, the authors need to do more than bioinformatically assess using a single programme. Ideally this should be experimentally verified or the conclusions appropriately qualified.

What is "control" in Fig. 1e? This is not mentioned in the text.

Legend Figure 2c. "For each gene, data were normalized to the mRNA level of mock transfected ESCs."

I thought this was endogenous levels of Trincr1.

Legend Fig 5d not sure if they are referring to T5 instead of T6 as there is no T6 in the figure.

Response to Reviewer Comments:

We sincerely thank the three reviewers for their careful analysis and constructive comments and suggestions for improving the manuscript. Please find below our point-by-point responses to reviewers' comments (texts in light blue)

Reviewer #1 (Remarks to the Author):

In this manuscript, the authors identified a TRIM71-interacting long noncoding RNA, which they named *Trincr1*, that plays an important role in FGF/ERK signaling and progenitor cell self-renewal. They showed that *Trincr1* is bound by Trim71 in mouse embryonic stem cells (ESCs), via its 5'-region and the NHL domain in Trim71. It negatively regulates ERK signaling, and is required for ESC identity and supports ESC self-renewal in sub-optimal culture conditions. *Trincr1* likely functions in the same pathway with Trim71, as Trim71 depletion rescues the defects of *Trincr1* knockout ESCs. Finally, they showed that *Trincr1* also represses FGF/ERK signaling in neural stem cells (NSCs) and is a negative regulator of NSC maintenance. Together, this study uncovered a novel regulator *Trincr1* in FGF/ERK signaling and highlights lncRNAs as important players in stem cell fate decisions.

While the role of Trim71 and *Shc1* in ERK signaling and ESC maintenance has been previously reported, the identification of *Trincr1* as well as its functional interaction with Trim71 are novel findings. To further strengthen the manuscript, the authors are encouraged to address the following questions and comments:

Based on the proposed model, *Trincr1* and Trim71 play important roles in FGF/ERK signaling. However, many of the experiments in the manuscript were carried out in culture conditions in which ERK signaling was blocked by PD0325901. The addition of the inhibitor may mask or undermine some of the observations, and it is therefore important to repeat some of the key experiments in culture conditions where FGF/ERK signaling is not inhibited (such as serum+LIF).

Answer: We thank the reviewer for this important suggestion. We now included analysis of pERK/ERK in serum+LIF condition. Consistently, we observed the upregulation of ERK signaling in *Trincr1*^{-/-} ESCs and the rescue of pERK level upon knocking down of Trim71 (**Figure R1**, also **Supplementary Figure 6a**).

Figure R1. *Trincr1* and *Trim71* opposingly regulate ERK activity in conventional ESC media in the absence of ERK inhibitor. Left, representative western blotting images; Right, quantification of pERK/ERK level in wild type, *Trincr1*^{-/-} ESCs treated with control shRNA vectors and *Trincr1*^{-/-} ESCs treated with two different shRNA vectors knocking down *Trim71*. Data were normalized to wild type ESCs treated with control shRNAs. Shown are mean \pm SD, n = 3. P values were calculated by paired two sided Student's *t*-test.

To provide further support for the proposed mechanism, it will be important to test the effect of *Trim71* overexpression in *Trincr1*^{-/-} ESCs, the effect of *Trincr1* KO on *Trim71* expression, and the effect of *Trincr1* KO on the interaction between TRIM71 and SHCBP1.

Answer: For the effect of *Trim71* overexpression in *Trincr1*^{-/-} ESCs, since *Trim71* and *Trincr1* play opposite function in ERK signaling, we guess the reviewer meant "the effect of *Trim71* overexpression in *Trincr1* overexpression ESCs". As shown below, *Trim71* overexpression successfully rescued the pERK level in *Trincr1* overexpressing cells (**Figure R2**). Together with data showing that *Trim71* knocking down rescued the pERK upregulation in *Trincr1*^{-/-} ESCs (**Figure R1** and **Figure 6b, c**), these results suggest that *Trim71* and *Trincr1* function in the same pathway to opposingly regulate ERK signaling.

Figure R2. *Trim71* overexpression blocked the effect of *Trincr1* overexpression in repressing ERK signaling. Left, western blotting image; Right, the overexpression level of *Trim71* and *Trincr1* in *Trincr1* and *Trim71* overexpressing ESCs. Control is the ESC transfected with empty overexpression vectors.

For the effect of *Trincr1* KO on *Trim71* expression, we performed RT-qPCR analysis for *Trim71* RNA and western blotting analysis for TRIM71 protein level in wild type,

Trincr1 KO and Trincr1 OE ESCs (**Figure R3**, also in **Figure 6g, h**). The conclusion is that the expression of Trincr1 (either KO or OE) does not affect the RNA and protein level of Trim71.

Figure R3. Trincr1 does not regulate the expression of Trim71 mRNA and protein. **a)** RT-qPCR analysis of Trim71 mRNA level in wild type, *Trincr1*^{-/-} and Trincr1 overexpression ESCs. The β -actin gene was used as a control. Data were normalized to wild type ESCs. Shown are mean \pm SD, n = 3. **b)** Western blotting analysis of TRIM71 protein level in wild type, *Trincr1*^{-/-} and Trincr1 overexpression ESCs. Left, representative western blotting image; Right, quantification of western blotting data. GAPDH was used as a control. Data were normalized to wild type ESCs. Shown are mean \pm SD, n = 3.

For the effect of Trincr1 KO on the interaction between TRIM71 and SHCBP1, we were not able to pull down SHCBP1 with endogenous FLAG-TRIM71 (FLAG knock in at the C terminus of TRIM71) in both wild type and *Trincr1* KO ESCs. This may be due to the detection limit of pull down and western blotting analysis. To circumvent around this issue, we overexpressed FLAG tagged TRIM71 and HA tagged SHCBP1 in 293T cell with or without Trincr1 overexpression and use FLAG antibody to IP TRIM71 and HA tagged SHCBP1. The results showed that Trincr1 overexpression can indeed weaken the interaction between TRIM71 and SHCBP1 (**Figure R4**, also in **Figure 6i**). The description for the data is included on Page 14 (highlighted in yellow).

Figure R4. Trincr1 weakens the interaction between TRIM71 and SHCBP1. IP with FLAG antibody was performed in HEK293 cells expressing HA-SHCBP1 and FLAG-TRIM71 with or without Trincr1 overexpression. Data were normalized to samples without Trincr1 overexpression. Shown are mean \pm SD, n = 3. P value was calculated by paired two sided Student's t-test.

Trim71 has been shown to promote ESC proliferation (PMID: 22735451). Based on this study, *Trincr1* represses Trim71. However, *Trincr1* also promotes ESC maintenance. Can the authors provide some explanations to reconcile these findings?

Answer: We thank the reviewer to raise this insightful question. We think that Trim71 has two independent functions in ESCs, to promote proliferation by repressing *Cdkn1a* (as shown in PMID: 22735451) and to promote ERK signaling by stabilizing SHCBP1. Only the latter is restrained by *Trincr1* to allow optimal self-renewal of ESCs. Indeed, our RNA-seq data shows that *Cdkn1a* expression was not affected by *Trincr1* knockout. (**Figure R5**).

Figure R5. *Cdkn1a* expression in wild type and *Trincr1* knockout ESCs.

Minor points:

Fig 1A, 1B: Consider moving them to the supplemental figures, as these observations do not promote the main conclusions in the paper and they have been reported previously in neural progenitor cells (PMID 22508726).

Answer: We agree with the reviewer and moved Fig 1A to the supplemental file (**Supplementary Figure 1b**) in the revised version. Fig 1B is removed as we added results from ESCs with Trim71 and Shcbp1 knocked down by CRISPRi approaches side by side (**Supplementary Fig. 1f-h**).

Fig 1D, 1E: Can the authors provide browser shots and RIP-qPCR data for additional Trim71-bound RNAs? There is a clear difference in fold-enrichment for *Trincr1* between RIP-seq and RIP-qPCR (~70 vs ~6).

Answer: Browser shots (**Figure R6, also Supplementary Figure 1i**) and RIP-qPCR data (**Figure R7, also Supplementary Figure 1j**) for additional Trim71-bound RNAs are shown below. We agree with the reviewer that there are clear differences in calculated enrichment score between RIP-seq and RIP-qPCR. This may be explained by different formula used to calculate the fold-enrichment in two experiments. For RIP-seq, fold-enrichment = FPKM in FLAG RIP samples versus FPKM in input samples, the input samples are total RNA from same FLAG-TRIM71 ESCs, this is a common practice for RIP-seq (similar in ChIP-Seq) since control RIP samples have very little

RNA and are not suitable for sequencing analysis; for RIP-qPCR, fold-enrichment= FLAG RIP RNA from FLAG-TRIM71 ESCs versus control ESCs transfected with empty FLAG vectors.

Figure R6. Representative browser shots for other Trim71 bound RNAs including H2afz, Atp5k and 5430416N02Rik.

Figure R7. RIP-seq data and qPCR data for candidate Trim71-bound RNAs. 7 out of 8 randomly chosen (different range of enrichment) potential targets based on RIP-seq data are validated. Data were normalized to ESCs transfected with control vectors. Shown are mean \pm SD, n = 3. P values were calculated by unpaired two-sided Student's *t*-test.

Fig 2G: How much fold-increase was Trincr1 overexpression compared to the endogenous level?

Answer: In Fig 2G (old version), the overexpression level of Trincr1 is shown below. It is about 1000 fold increase for the short isoform and 2700 fold increase for the long isoform of Trincr1 compared to the endogenous level (**Figure R8**, also **Supplementary Figure 2e**). During revision, we obtained ESCs overexpressing Trincr1 at much lower level and repeated the experiment (**Figure R9**, also **Supplementary Figure 2f**). Inhibition of ERK signaling is also observed in these ESCs. We were not able to identify Trincr1 overexpressing ESC colonies with even lower expression of Trincr1.

Therefore we can not conclude about exact amount of Trincr1 needed to repress Trim71 in ESCs. However, our results from *Trincr1* knockout experiments suggest that endogenous level of Trincr1 has a significant inhibitory effect on Trim71.

Figure R8. Trincr1 overexpression level in Figure 2G (old version), Figure 2f (revision).

Figure R9. Low level overexpression of Trincr1 short isoform represses ERK signaling.

Fig 3C, 3D: Consider removing 3C, as it presents the same information as 3D.

Answer: We moved Figure 3C to Supplementary Figure. 3e.

Fig 5H: FL-Trincr1 RIP-seq signal appears to be much lower than what was in Fig 1E. Interestingly, delta-RB mutant appears to show stronger interaction with Trincr1. Can the authors comment on these observations?

Answer: Both Fig 5H and Fig 1E were results from RIP-qPCR, we think the difference is likely caused by experiment variation. In the revised manuscript, we repeated FL-TRIM71 RIP experiments 2 more times (independent experiments) and made a new figure to replace Fig. 1E. As shown in **Figure R10**, the fold enrichment value was similar to that shown in Fig. 5H (Supplementary Figure 5e in the revised manuscript). For the second question, we repeated experiments with FL- and delta-RB mutant side by side independently for 4 times. In agreement with Fig. 5H, delta-RB mutant always showed stronger interaction with Trincr1 than FL-Trim71 (**Figure R11, also in Figure 5h in the revised manuscript**). We think that the RB domain (often interacting with other proteins) may affect the accessibility of NHL domain to RNA. Structural studies

of Trim71 should be able to provide further insights on this difference. Interestingly, similar intramolecular inhibition of RNA binding was reported for several other proteins. In a recent study (<https://academic.oup.com/nar/article/46/12/5894/5003453>), the full-length TLS/FUS has weaker binding to ncRNA TERRA than its RGG3 and RBD domains, indicating that other parts of the protein could inhibit the binding of TLS/FUS to ncRNA TERRA. Similar to TLS/FUS study, another paper published in 2012 (<https://www.nature.com/articles/ncomms2005>) showed that RNA binding ability of NXF1 is inhibited by its own NTF2L domain.

Figure R10. RIP qRT-PCR of full length TRIM71. Data were normalized to ESCs transfected with control vectors. Shown are mean \pm SD, n = 4. P value was calculated by paired two-sided Student's *t*-test.

Figure R11. RIP qRT-PCR of full length and delta-RB mutant TRIM71. Data were normalized to ESCs transfected with control vectors. Shown are mean \pm SD, n = 4. P values were calculated by paired two-sided Student's *t*-test.

Fig 6B, 6C: There is some visual inconsistency in pERK/ERK between the blot in 6B and the quantitation in 6C for Trim71-shRNA2: sh2 appears to reduce pERK level more substantially than sh1 in 6B, but that is not the case in 6C.

Answer: Thank the reviewer for pointing out this issue. Figure 6C showed the average of four independent experiments (**Figure R12**). Figure 6B was shown as a representative image. In the revised version, we replaced Figure 6B with a more representative image (Up right of Figure R12).

Figure R12. Western blotting analysis of pERK. Shown are data from four independent experiments for wild type ESCs treated with control shRNAs and *Trincr1*^{-/-} ESCs treated control or Trim71 shRNAs. Data were normalized to wild type ESCs treated with control shRNAs. Figure panels used in the old version and revised version are indicated.

Fig 6: Will disruption of the NHL-domain in Trim71 also rescue the *Trincr1* deletion phenotype?

Answer: We think it will, at least partially. Data from an ongoing project in our lab showed that the interaction between TRIM71 and SHCBP1 is partially dependent on NHL-domain (**Figure R13**). Therefore, the deletion of NHL domain will likely diminish its activity in promoting ERK signaling. In other words, NHL deletion may generate a phenotype similar to knocking down Trim71, which we have shown to be able to rescue the *Trincr1* deletion phenotype.

Figure R13. Co-IP of SHCBP1 with full length or NHL domain deleted TRIM71. Data were normalized to samples expressing full length TRIM71. Shown are mean \pm SD, n = 3. P value was calculated by paired two-sided Student's *t*-test.

Fig 7: The authors referred to the ESC derived neural stem cells as GPCs. However, they are really NSCs based on a previous report (PMID: 16086633) and the fact that they express neural progenitor markers Nestin and Pax6. Need to clarify the nomenclature.

Answer: Thank the reviewer for pointing this out. We checked literature carefully and agree with the reviewer. Neural progenitor/stem cells are more appropriate to be used in this case. We fixed it by using neural progenitor cells in the revised manuscript.

Fig 7A: “The result is as expected since these cells does not express any Trim71”. Can the author compare the expression of Trim71 between 3T3 and ESCs?

Answer: We included qPCR data for Trim71 in 3T3 and ESCs (**Figure R14, also as Supplementary Figure 7a**). Trim71 expression is ~1000 fold lower in 3T3 cells compared to ESCs.

Figure R14. qPCR for Trim71 in 3T3 and ESCs. Data were normalized to β -actin and then to ESCs. Shown are mean \pm SD, $n = 3$. The right panel shows representative amplification plots in 3T3 and ESCs.

Fig 8: Consider changing ESC to Progenitor Cells, so that the model can encompass results from both ESCs and NSCs.

Answer: Thank the reviewer to raise this issue. We removed this figure based on the suggestion from the second reviewer. We are continuing the work on Trim71 and Trincr1 related function and mechanisms. Hopefully we can provide a more comprehensive model in near future.

Other wording changes: Thank the reviewer for suggestion on these wording changes. We agree with all of them.

1. “Together, these data show that similar to its role in neural progenitor cells, Trim71 promotes FGF/ERK signaling through upregulating SHCBP1 protein in ESCs”. The causality has not been fully established at this point.

Answer: We changed this sentence to "Together, these data show that similar to its role in neural progenitor cells, Trim71 could promote FGF/ERK signaling in ESCs, likely through upregulating SHCBP1 protein." in the revised version.

2. “qRT-PCR analysis for a panel of pluripotency markers and early differentiation marker Fgf5 confirmed that Trincr1 is important for the self-renewal and pluripotency of ESCs in PD+LIF”. Marker analysis only shows the requirement for Trincr1 in the maintenance of ESC identity, not necessarily in self-renewal and pluripotency.

Answer: We changed this sentence to "qRT-PCR analysis for a panel of pluripotency markers and early differentiation marker Fgf5 confirmed that Trincr1 is important for the maintenance of ESC identity in PD+LIF" in the revised version.

3. Similarly, “Together, these data demonstrate that Trincr1 is important for the self-renewal and pluripotency of ESCs cultured in various sub-optimal conditions”. The data showed that Trincr1 is important for self-renewal, but did not provide sufficient information on its role in pluripotency.

Answer: We changed this sentence to "Together, these data demonstrate that Trincr1 is important for the self-renewal of ESCs cultured in various sub-optimal conditions" in the revised manuscript.

Reviewer #2 (Remarks to the Author):

Li et al present an interesting new study describing a novel role for the lncRNA TRINCR in fine-tuning cellular responses to FGF signaling in ESCs. They use a wide variety of appropriate molecular approaches to identify a putative (though not necessarily direct) interaction between TRINCR and TRIM71 – a known modulator of FGFR activity. The study builds in a logical fashion, starting with the identification of a modest effect of TRINCR knockdown on ERK signaling, then using RIP-seq to identify TRIM71 as a potential binding partner.

The introduction is rather brief - no discussion of TRIM71 – this could be expanded upon, rather than just coming up in the discussion.

Answer: We thank the reviewer for the suggestion to improve the readability of the manuscript. We included the discussion for Trim71 in the introduction part in the revised version (Page 4-5, highlighted in yellow).

One weakness of the manuscript is the failure to display direct binding between TRINCR and TRIM71 – they need to mix recombinant protein with IVT RNA and show that the two interact.

Answer: We presented data showing that IVT Trincr1 can pull down TRIM71 protein from cell extracts and that antibody to FLAG-TRIM71 can enrich Trincr1. In addition, we showed that deletion of RNA binding domain NHL completely abolished the interaction between Trincr1 and Trim71, suggesting their interaction dependent on NHL domain. However, as pointed out by the reviewer, these results do not exclude the possibility that the interaction between Trincr1 and TRIM71 is mediated by other

proteins or RNAs. Therefore, the electromobility shift assay (EMSA) suggested by the reviewer is required to test whether the interaction is direct or not. The EMSA requires recombinant protein TRIM71. Unfortunately, we were not able to purify recombinant TRIM71 from bacteria, preventing us from performing EMSA experiments. For this reason, we added a few sentences in the discussion to discuss the possibility that the interaction between Trincr1 and TRIM71 could be indirect and dependent on other factors (Page 16).

Taking each of the figures in turn

Fig 1

Why do they use FGF2 and not the more relevant FGF4?

Answer: We used bFGF (FGF2) since bFGF is often used to induce the exit of naive pluripotency and to convert ESCs into EpiSCs or EpiSC-like cells (e.g. Klf4 reverts developmentally programmed restriction of ground state pluripotency, PMID: 19224983; Reconstitution of the mouse germ cell specification pathway in culture by pluripotent stem cells, PMID: 21820164). Furthermore, to address this concern, we repeated ERK inhibition experiments with FGF4. As shown below (**Figure R15**), Trincr1 overexpression similarly inhibited the phosphorylation of ERK induced by bFGF and FGF4.

Figure R15. Overexpression of Trincr1 short isoform represses ERK phosphorylation induced by FGF2 or FGF4. Left, representative images for western blotting; right, quantification of western blotting data. Data were normalized to ESCs transfected with control overexpression vectors in each condition. Shown are mean \pm SD, $n = 3$. P values were calculated by paired two-sided Student's *t*-test.

The argument for quantitating ERK and P-ERK on different blots is very odd – not one I've ever come across in signaling papers and when the differences are so minor it would be nice to see data from the same blot.

Answer: To quantitate ERK and P-ERK on the same blot requires stripping of the blot which might affect the quantification of the second antibody. Instead, we used same protein samples to run two gels and performed blotting separately. This also saved time for us for avoiding stripping step. In addition, to address this concern, we performed blotting as the reviewer suggested and similar results were obtained by using different blots or the same blot on protein samples from the same preparation (**Figure R16**).

Figure R16. Quantification of ERK and pERK by different blots (left panel) or the same blot (right panel) in *Trincr1* overexpressing NPCs. Data were normalized to wild type NPCs. Protein samples from the same preparation were used.

They also chop and change between quantitating with numbers and showing graphs of quantitation. In (b) there is no mention of the vast difference between lanes 1+4 compared to (a) lanes 1+3; these should be very similar as they are the exact same conditions. No quantitation in (b) and $n=2$ – this is a common problem throughout – sometimes $n=2$, sometimes 3.

Answer: Thank the reviewer to point this out. The purpose of these experiments was to confirm findings from previous findings. In the revision, we included results from Crispr ESCs with 2 different sgRNAs designed for both *Trim71* and *Shcbp1* (**Figure R17**). The data are from 3 independent experiments and are shown in **Supplementary Figure 1f-h** in the revised version. The description for the data is included on page 6 (highlighted in yellow). We also fixed other n problems by repeating experiments and made sure all key results are now with $n \geq 3$. Finally, based on the first reviewer's suggestion that the above information "do not promote the main conclusions in the paper and they have been reported previously in neural progenitor cells". We placed the old Figure 1a and newly added Crispr results in Supplementary Fig. 1b, 1f-h in the revised version, respectively. Figure 1b is removed. There must be some errors with lane 1 or 4 as they should express similar level of SHCBP1 (**Figure R18**).

Figure R17. Western blotting analysis of SHCBP1 and pERK in *Trim71* and *Shcbp1* knockdown ESCs cultured in 2i+LIF or CHIR induced by bFGF. **a)** RT-qPCR showing knock down efficiency of CRISPRi constructs targeting *Trim71* and *Shcbp1*. **b)** Representative images for western blotting analysis of pERK and ERK. **c)** Quantification of data in b. Data were normalized to ESCs treated with control gRNAs. Shown are mean \pm SD, $n = 3$. P values were calculated by paired two-sided Student's *t*-test.

Figure R18. Western blotting analysis of SHCBP1 protein in ESCs in 2i+LIF or CHIR+bFGF condition. Right panel shows the quantification. Data were normalized to 2i+LIF. Shown are mean \pm SD, n = 3.

Fig 2

(d) blots are Chir+FGF2 not 2i+LIF. The differences between Sh1+2 show why it is important to look at more than one (unlike in Sup Fig 1(a))

Answer: Sorry for the confusing part. (d) blots were meant to show that the background pERK level is increased in *Trincr1* knocking down ESCs in 2i+Lif conditions. To address concerns for one shRNA for *Trim71*, we included results from two more CRISPRi mediated *Trim71* knocking down ESCs (**Figure R17** above). The old figure 2d is moved to supplementary figure 2c in the revised manuscript.

Fig 3

Why does SLE appear in both up and down regulated in (e)?

Answer: This is likely due to the intrinsic limitation of pathway analysis which is only based on statistics. Based on our data, SLE is indeed enriched in both up and downregulated genes. However, we noticed these genes are generally lowly expressed (average FPKM~0.8), therefore the results are likely due to the large variation in the expression of lowly expressed genes. In the revised manuscript, we performed analysis only for genes with average FPKM \geq 1. Results are as follows (**Figure R19**). In the revised manuscript, enrichment for upregulated genes is shown in **Figure 3d** and downregulated genes in **Supplementary Figure 3f**.

Figure R19. KEGG pathway analysis of differentially expressed genes between *Trincr1*^{-/-} and wild type ESCs. Left, upregulated genes; Right, downregulated genes. Fold of enrichment and -Log₂ (P value) are shown.

Fig 4

(c) needs controls for fractionation efficiency. Nfx1 must be repeated to get n=3

Answer: We included controls for fractionation efficiency in the revised manuscript (Figure 4c, also see Figure R20 below); Nfx1 is repeated to get n = 3.

Figure R20. Fraction of Trincr1 in cytoplasm and nucleus in CRISPRi ESCs. Shown are mean \pm SD, n = 3. P values were calculated by unpaired two sided Student's *t*-test.

Fig 5

(b) needs a graph of quantification and data for T5.

Answer: We included quantification in Figure 5b and data for T5 in the revised manuscript (Supplementary Figure 5b and also Figure R21).

Figure R21. Western blotting analysis of phosphorylated ERK in control and Trincr1_S overexpressing ESCs induced by bFGF. Data were normalized to ESCs treated with control overexpression vectors. Shown are mean \pm SD, n = 3. P values were calculated by paired two-sided Student's *t*-test.

F+h) colour schemes are very confusing – unnecessary.

Answer: We changed color of F. Thanks for pointing this out.

Need to comment on why deltaRB gives more TRINCR1 enrichment compared to FL.

Answer: We think that the RB domain (often interacting with other proteins) may affect the accessibility of NHL domain to RNA. Structural studies of Trim71 in future may be able to provide further insights on this difference. Interestingly, similar intramolecular inhibition of RNA binding was reported for several other proteins. In a study published in July (<https://academic.oup.com/nar/article/46/12/5894/5003453>), the full-length TLS/FUS has weaker binding to ncRNA TERRA than its RGG3 and RBD domains, indicating that other parts of the protein could inhibit the binding of

TLS/FUS to ncRNA TERRA. Similar to TLS/FUS study, another paper published in 2012 (<https://www.nature.com/articles/ncomms2005>) showed that RNA binding ability of NXF1 is inhibited by its own NTF2L domain.

D) T5 not T6 in legend.

Answer: Corrected as T5. Thanks!

Fig 6

a) Explain why *Trincr1*^{-/-} has less Trim71 than WT.

Answer: There was a small but statistically significant decrease in Trim71 RNA level in *Trincr1*^{-/-} versus wild type ESC in the old Figure 6a. In the revised manuscript, we performed more careful analysis on the effects of *Trincr1* knockout and overexpression on Trim71. We performed RT-qPCR analysis for Trim71 RNA and western blotting analysis for TRIM71 protein level in wild type, *Trincr1* KO and *Trincr1* OE ESCs (**Figure R22**, also **Figure 6g, 6h**; **Figure R23**, also **Figure 6a**). The conclusion is that the expression of *Trincr1* (either KO or OE) does not affect the RNA and protein level of Trim71.

Figure R22. *Trincr1* does not regulate the expression of Trim71 mRNA and protein. **a)** RT-qPCR analysis of Trim71 mRNA level in wild type, *Trincr1*^{-/-} and *Trincr1* overexpression ESCs. The β -actin gene was used as a control. Data were normalized to wild type ESCs. Shown are mean \pm SD, n = 3. **b)** Western blotting analysis of TRIM71 protein level in wild type, *Trincr1*^{-/-} and *Trincr1* overexpression ESCs. Left, representative western blotting images; Right, quantification of western blotting data. GAPDH was used as a control. Data were normalized to wild type ESCs. Shown are mean \pm SD, n = 3.

Figure R23. Knocking down Trim71 in *Trincr1*^{-/-} ESCs. Left is the old version figure 6a; Right is the revised version of figure 6a that is more representative for the expression of Trim71 in *Trincr1*^{-/-} ESCs. P values were calculated by unpaired two-sided Student's *t*-test.

Fig 7

They need to present RTPCR evidence to back up their claim that 3T3 cells don't express Trim71 and GPCs do. Could do with more informative labeling in (a). c) is very uninformative – for a differentiation marker would be better

Answer: We included qPCR data for Trim71 in 3T3, NPCs and ESCs in the revised version (Supplementary Figure 7a, also Figure R24). We modified our labeling in Supplementary Figure 7c (the old Figure 7a) by including quantification. We moved the old Figure 7c to supporting information as Supplementary Figure 7d and added immunofluorescence staining data for Nestin (Figure 7c, also Figure R25).

Figure R24. qRT-PCR analysis of Trim71 expression in NPC, 3T3 and mouse ESCs. The β -actin gene was used as a control. Data were normalized to NPCs. Shown are mean \pm SD, n = 3.

Figure R25. Immunofluorescence staining for NPC marker Nestin in wild type and Trincr1_S overexpressing NPCs. Scale bars, 100 μ m.

Fig 8

Not very attractive figure. They don't show any evidence of action on FGFR2 directly throughout the paper and no evidence of direct interaction between Trincr1 and TRIM71. No mention of Ub elsewhere – need to test this with proteasome inhibitors.

Answer: We removed Figure 8 in the revised version as suggested. We are doing more investigations and hopefully can provide a more comprehensive model in future.

Sup Fig 1

Need to test more than one ShRNA.

Answer: The results of these experiments were also supported by previously published results from other groups. In addition, we included results from two siRNAs as shown in **Supplementary Figure 1c, 1d**. We also added CRISPRi results (2 different gRNAs for each gene, **Figure R26**, also **Supplementary Figure 1f-h**) in the revised version.

Figure R26. ERK signaling is impaired in Trim71 and Shcbp1 knocking down ESCs. a) RT-qPCR showing knock down efficiency of CRISPRi constructs targeting Trim71

and Shcbp1. The β -actin gene was used as a control. Data were normalized to ESCs treated with control gRNAs. Shown are mean \pm SD, n = 3. **b)** Western blotting analysis of SHCBP1 and phosphorylated ERK in Trim71 and Shcbp1 knockdown ESCs cultured in 2i+LIF or CHIR+bFGF. **c)** Quantification of **b**. Data were normalized to ESCs treated with control gRNAs. Shown are mean \pm SD, n = 3. P values were calculated by paired two-sided Student's *t*-test.

D) cannot be true – how to go from 200,000 to 5 million cells in 2 days. They have only measured one timepoint so why the line graph?

Answer: Sorry for this labeling error. It should be 58 hours. We corrected this and made bar graph using data from another batch of experiments as requested (**Figure R27**, also **Supplementary Figure 1e**).

Figure R27. Knocking down Trim71 impairs the proliferation of ESCs. Shown are cell numbers for ESCs transfected with control shRNA vectors or shRNA vectors against Trim71. Shown are mean \pm SD, n = 3. P values were calculated by unpaired two-sided Student's *t*-test. Population doubling time is calculated and shown at right. Similar results were shown by Gregory et al (PMID: 22735451).

Throughout paper all blots need kDa

Answer: We included kDa for all blots in the revised version

Sup Fig 3

Why only n=2? What is purple dotted line in (a)? Scale bars in (b). Error bars in (d). Stain in d is what?

Answer: For the first question, it was a small screen for identifying at which conditions *Trincr1*^{-/-} has the most dramatic effects on the expression of pluripotency genes (32 samples in total analyzed). We repeated some conditions such as PD+LIF in **Figure 3a** (n = 3). In addition, RNA-seq experiments were performed for 2i+LIF and PD+LIF conditions (**Figure 3c**). For the second question, purple dotted line means 0.7, it was an arbitrary cut off to choose a culture condition with the most significant effect to follow up. For the third question, we added scale bars in revised version. In addition, we added error bars in (d). Stain in d is alkaline phosphatase staining which serves as a marker for pluripotent stem cells. We added this information in the figure legend.

Sup Fig 4

How is the MEK pathway 1.8x upregulated – not according to the figure?

Answer: It was 1.4x upregulated. We corrected this in the revised version. Thank the reviewer for pointing out this mistake.

Reviewer #3 (Remarks to the Author):

In this manuscript, the authors identified a lincRNA that binds to TRIM71 and has a potential role in maintaining ESCs pluripotency in the context of FGF/ERK signalling. It has been reported previously that TRIM71 promotes FGF/ERK signalling by stabilizing SHCBP1. In the manuscript, the authors showed that the knockdown of Trim71 by shRNA and siRNA results in a reduction of p-ERK and the expression of SHCBP1. They also found that TRIM71 binds at least three lincRNAs and explored in more detail the interaction with Trincr1. They show that TRIM71 NHL domain interacts with the 1-140nt of Trincr1 and that it is exported into the cytoplasm by ThOC complex. Overall, the authors present some reasonable evidence to suggest a role for Trincr1 in controlling ERK activity levels in ES cells. However, as a lot of the effects are seen when ERK signalling is blocked through MEK inhibition, it is possible that Trincr1 works via a different route. Hence there there are a number of areas that require further attention.

Major issues:

(1) Throughout the paper, the effects on ERK phosphorylation are difficult to make out. The authors do provide quantification but this is from a single repeat. As the experiments have presumably been repeated multiple times, the authors should provide averages of these experiments (with associated error ranges).

Answer: This is an important suggestion. We repeated all key experiments ($n \geq 3$) and provided average with error bars (mean \pm SD) as suggested in the revised manuscript.

(2) In Figure 1, the authors show a reduction on Trim71 mRNA and these leads to a reduction in the levels of SHCBP1. The knock down of Trim71 and Shcbp1 result in a reduction of the levels of p-ERK. However, the authors do not address what effect does this has on pluripotency and self-renewal. I think this is important for assessing the effect of Trincr1 in this pathway.

Answer: This is a very important suggestion. In the revised manuscript, we tested colony formation ability of Trim71 and Shcbp1 knocking down ESCs in N2B27+LIF condition and found that both Trim71 and Shcbp1 knocking down can promote ESC self-renewal in N2B27+LIF condition, consistent with its role in ERK signaling (**Figure R28**).The data is shown in **Figure 1a** and the description for the data is included on Page 6 (highlighted in yellow).

Figure R28. Knocking down Trim71 or Shcbp1 promotes ESC self-renewal in N2B27 supplemented with LIF alone. Shown are results from colony formation assay for ESCs treated with control, Trim71 and Shcbp1 CRISPRi gRNAs. Data were normalized to ESCs treated with control gRNAs grown in 2i+LIF condition. Shown are mean \pm SD, $n = 3$. P values were calculated by unpaired two-sided Student's t -test.

(3) In Fig. 1b, virtual complete loss of SHCBP1 has a similar effect on P-ERK levels compared to <50% depletion in Fig. 1a. Why is this?

Answer: Thanks for pointing this out. In the revised manuscript, we added CRISPRi results (**Supplementary Figure 1f-h**) and confirmed that the phenotype is real when western blotting is performed side by side. This could be explained by that TRIM71 regulates ERK pathway through multiple factors besides SHCBP1.

(4) Did the authors run controls for Fig. 1c? ie were replicates run? Was an IP from the parental cell line done? Some validation of Trincr1 is done but none of the others are validated. Therefore if no replicates are done etc, then it should be made clear that all the other bound RNAs should be considered preliminary and Fig. S1f is essentially meaningless.

Answer: For RIP-seq experiments, two biological replicates were performed. In addition, the RIP seq was done with control cell lines (ESCs transfected with empty vectors instead of FLAG-TRIM71 vectors). Since control ESCs do not express any FLAG-TRIM71, only very few nonspecific binding RNAs were pulled down by FLAG antibodies. For this reason, we did not normalize the data to the control cell line, instead to the input sample as many other studies did. For validation, we added validation for another 8 randomly chosen TRIM71 bound candidate RNAs (enrichment fold ranges from ~5 to 140 fold), 8 out of 9 (counting Trincr1) candidates were validated (**Figure R29, R30**). Nevertheless, we agree with the reviewer that the results for these candidate bound RNAs are preliminary, therefore removed Fig. S1f as requested. We are currently developing more robust approaches such as CLIP to identify Trim71 interacting RNA targets.

Figure R29. Representative browser shots for other Trim71 bound RNAs including H2afz, Atp5k and 5430416N02Rik.

Figure R30. RIP-seq data and qPCR data for candidate TRIM71 binding RNAs. 7 out of 8 randomly chosen potential targets based on RIP-seq data were validated. Data were normalized to ESCs treated with control overexpression vectors. Shown are mean \pm SD, n = 3. P values were calculated by unpaired two-sided Student's *t*-test.

(5) In Fig. 1e, rather than using n=2, of presumably technical replicates, the authors should provide the average of the 3 independent experiments they refer to. Note that statistics cannot be done on n=2.

Answer: We provided the average of 4 independent experiments with standard deviation in the revised version (**Figure 1d**).

(6) The authors show that Trincr1 binds TRIM71 and that differentiating cells show a reduction of Trincr1 expression (others have also shown a reduction in TRIM71 expression upon differentiation). However, the levels of TRIM71 during differentiation are not assessed. I think that if Trincr1 is having an effect on differentiation through TRIM71 it would be important to show the levels of the protein also vary. (In previous

reports it has been shown that the levels of TRIM71 are reduced upon differentiation of Neural progenitor cells).

Answer: Thank the reviewer for this important suggestion. We included expression level of Trim71 during retinoid acid (RA) differentiation in the revised version. As shown in **Figure R31** (also **Figure 2c** in the revised manuscript), Trim71 was not significantly changed during 4 days of RA induced differentiation, while Trincr1 expression decreased at early days of differentiation. These data suggest that the decrease of Trincr1 at early time of differentiation releases its block of Trim71 activity.

Figure R31. Trim71 and Trincr1 expression during RA differentiation. The Gapdh gene was used as a control. Data were normalized to undifferentiated ESCs (D0). Shown are mean \pm SD, $n = 3$. P values were calculated by unpaired two-sided Student's *t*-test.

In Fig. 2f, the authors show that P-ERK is detectable in 2i+Lif conditions. However this should not be possible due to the MEK inhibitor being present. Indeed this is what the authors themselves show (see Fig. 1a). Please explain. The same applies to Fig. 1d. Also, how can increases in P-ERK levels be seen in the presence of PD inhibitor, as there should be no activation from MEK present. Similarly in Fig. 3a, Trincr1 has an effect but MEK (and hence ERK) will be inhibited under these conditions. This suggests that Trincr1 might be working through an alternative route to the ERK signalling pathway.

Answer: The small molecule PD0325901 may only inhibit but not completely abolish the activity of MEK. The remaining background activity of pERK could be very important for ESCs, since a recent study (PMID: 26483458) showed that complete knockout of Erk1/2 disrupts the genomic stability and self-renewal of ESCs. We are not the only group having detected pERK in 2i+LIF conditions. For example, Yamanaka group also detected pERK in 2i+LIF conditions (**Figure R32**). However, compared to pERK level in FGF induced condition, the background pERK level in 2i+LIF condition is indeed very low. As stated in our Methods section, for all experiments to determine background pERK level in 2i+LIF and PD+LIF condition, we used more sensitive methods for detection. Specifically, HRP-conjugated anti-rabbit secondary antibodies were used and membranes were incubated with the Western ECL Substrate for the detection of pERK bands. For all other experiments to determine pERK level upon FGF induction, fluorescent secondary antibodies were used. In

addition, 200 μ g cell extracts were used for HRP detection of background pERK level, while only 60 μ g cell extracts were loaded for fluorescent antibody experiments to detect FGF induced pERK. Furthermore, in the revised manuscript, we included analysis of pERK/ERK in serum+LIF condition in the absence of MEK inhibitors. Consistently, we observed the upregulation of ERK signaling in *Trincr1*^{-/-} ESCs and the rescue of pERK level upon knocking down of Trim71 (Figure R33, also Supplementary Figure 6a). These results strongly support our conclusion that *Trincr1* inhibits ERK signaling through TRIM71.

[Redacted]

Figure R32. pERK is detected in ESCs cultured in 2i+LIF condition by others. Shown is a figure cropped from Shinya Yamanaka (PMID: 28003464, Figure 2).

Figure R33. *Trincr1* and Trim71 opposingly regulate ERK activity in conventional ESC media in the absence of ERK inhibitor. Left, representative western blotting images. Right, quantification of pERK/ERK level in wild type, *Trincr1*^{-/-} ESCs treated with control shRNA vectors and *Trincr1*^{-/-} ESCs treated with two different shRNA vectors knocking down Trim71. Data were normalized to wild type ESCs treated with control shRNA vectors. Shown are mean \pm SD, n = 3. P values were calculated by paired two sided Student's *t*-test.

(7) If always working in the same way, why do different genes respond to *Trincr1*

depletion under different conditions in Supplementary Fig. 3a?

Answer: Pluripotency genes are regulated redundantly or collaboratively by multiple regulators and pathways, therefore the impact on the expression of specific pluripotency genes by manipulating a given factor (a gene or pathway) is often dependent on specific culture conditions. For example, as shown below by PMID: 23040478, knocking out an important pluripotency factor Esrrb affects the expression of Sox2 and Oct4 in CHIR condition, but not in LIF+PD or LIF+serum conditions (**Figure R34**). Other pluripotency genes such as Nr5a2 and Klf5 also respond differently to Esrrb knockout in different culture conditions.

[Redacted]

Figure R34. The impact of Esrrb knockout on pluripotency gene expression varies in different culture conditions. (From Figure 4 of PMID: 23040478)

(8) In figure 4, they show that Thoc5 is responsible for exporting Trincrl into the cytoplasm. They also show that the Thoc5 knockdown has an effect on phosphorylation of ERK but the control that they use is only mock transfected cells. It would be more

appropriate to use a non-targeting guide or even one of the other guides that had no effect on the levels of cytoplasmic Trincr1, in this way they can rule out any effect of the CRISPRi on p-ERK.

Answer: We are very sorry for not making it clear that the control in figure 4 is empty CRISPRi vector with 18 bp non-targeting guide. The guide sequence is 5'-GGGTCTTCGAGAAGACCT-3'. In addition, below we included a gRNA targeting Psme4 and L1td1 as the control. The results are consistent with that knocking down Thoc5, but not CRISPRi approach itself, led to the increase of pERK level (**Figure R35**).

Figure R35. Knocking down Thoc5 but not L1td1 or Psme4 increases pERK level in ESCs. Quantification of pERK/ERK is shown. Data were normalized to ESCs treated with control gRNAs.

Fig. 4 is peripheral to the main story and could be moved to the Supplementary data.

Answer: We think that Fig. 4 strengthens the main story in two levels: First, Trincr1 is mainly located in cytoplasm, consistent with its function in controlling the activity of cytoplasm located TRIM71; Second, the export of Trincr1 from nucleus to cytoplasm is regulated by Thoc5, a gene that previously has been shown to be important for ESC self-renewal through specifically regulating the transport of pluripotency mRNAs (PMID: 24315442), therefore consistent with the function of Trincr1 in ESC self-renewal, since genes with similar functions are generally thought to be regulated through similar mechanisms.

(9) The truncation experiments in figure 5b lack the levels of p-ERK in the T5 fragment, which should be similar to the ones of T2. I think it is important to show this as they use this fragment as control in further experiments.

Answer: Sorry for missing that. We added the figure for T5 in the revised version (**Figure R36**, also **Supplementary Figure 5b**).

Figure R36. Western blotting analysis of phosphorylated ERK in control and *Trincr1_S* overexpressing ESCs induced by bFGF. Data were normalized to ESCs treated with control overexpression vectors. Shown are mean \pm SD, $n = 3$. P values were calculated by paired two-sided Student's *t*-test.

(10) In figure 6a, the authors show that the levels of TRIM71 mRNA in *Trincr1* $-/-$ cells are reduced compared to wild-type cells. They also show an increase in the levels of SHCBP1 protein (but not mRNA) in the knockout cells (Fig. 6f). This is contrary to Figure 1 where they show that the levels of SHCBP1 go down when TRIM71 is downregulated. If *Trincr1* is a negative regulator of TRIM71, the levels of the proteins would need to be assessed to see that the effect they are observing is not independent from TRIM71.

Answer: These are also important suggestions. There was a small but statistically significant decrease in *Trim71* RNA level in *Trincr1* $-/-$ versus wild type ESC in the old Figure 6a. In the revised manuscript, we performed more careful analysis on the effects of *Trincr1* knockout and overexpression on *Trim71*. We performed RT-qPCR analysis for *Trim71* RNA and western blotting analysis for TRIM71 protein level in wild type, *Trincr1* KO and *Trincr1* OE ESCs (Figure R37, also Figure 6g, 6h; Figure R38, also Figure 6a). The conclusion is that the expression of *Trincr1* (either KO or OE) does not affect the RNA and protein level of *Trim71*.

Figure R37. *Trincr1* does not regulate the expression of *Trim71* mRNA and protein. a) RT-qPCR analysis of *Trim71* mRNA level in wild type, *Trincr1* $-/-$ and *Trincr1*

overexpression ESCs. The β -actin gene was used as a control. Data were normalized to wild type ESCs. Shown are mean \pm SD, n = 3. **b)** Western blotting analysis of TRIM71 protein level in wild type, *Trincr1*^{-/-} and *Trincr1* overexpression ESCs. Left, representative western blotting images; Right, quantification of western blotting data. GAPDH was used as a control. Data were normalized to wild type ESCs. Shown are mean \pm SD, n = 3.

Figure R38. Knocking down Trim71 in *Trincr1*^{-/-} ESCs. Left is the old version figure 6a; Right is the revised version of figure 6a that is more representative for the expression of Trim71 in *Trincr1*^{-/-} ESCs. Shown are Trim71 expression levels in wild type ESCs treated with control shRNAs, *Trincr1*^{-/-} ESCs treated with control shRNAs or shRNAs against Trim71. Data were normalized to wild type ESCs treated with control shRNA vectors. Shown are mean \pm SD, n = 3. P values were calculated by unpaired two-sided Student's *t*-test.

In figure 3a, they show that the loss of *Trincr1* diminished the expression of pluripotency genes. In contrast, RT-qPCR results in figure 7c show that markers for progenitor cells are significantly downregulated in GPCs overexpressing *Trincr1*. Is *Trincr1* having opposite functions in ESCs and GPCs?

Answer: *Trincr1* has the same function in terms of inhibiting ERK signaling. However, ESCs and GPCs (corrected as neural progenitor cells [NPCs] in the revised manuscript) have different requirements for ERK signaling. ERK signaling promotes the self-renewal of NPCs, while ERK inhibition promotes the self-renewal of ESCs. Therefore our results from both cell types are consistent with ERK inhibition function of *Trincr1*.

(11) What effect does *Trincr1* has on TRIM71? And how do they know that the effects they see are because of the binding of *Trincr1* to TRIM71?

Answer: We showed that overexpression or knockout of *Trincr1* does not affect the expression of Trim71 mRNA and protein (please also see our response for question 10 above, **Figure R37** and **R38**). Therefore, *Trincr1* does not regulate the expression level of Trim71 mRNA or protein. For the second question, we showed data that *Trincr1* can pull down TRIM71 protein and TRIM71 RIP enriched *Trincr1*. In addition, deletion of RNA binding domain NHL impairs the interaction between *Trincr1* and TRIM71.

Moreover, the fragment T5 of Trincr1 that does not bind TRIM71 has no function in ERK signaling, while the fragment T3 of Trincr1 that can bind TRIM71 inhibits ERK signaling. These data support that Trincr1 represses ERK signaling through the interaction between Trincr1 and TRIM71. To provide further evidence on the mechanism we are proposing, we are currently mapping the exact motif(s) on Trincr1 that interacts with TRIM71. Once mapped, the mutation of TRIM71 interacting motif(s) should abolish the ERK inhibition activity of Trincr1 if our hypothesis is right.

Minor comments:

To definitively state that trincr1 is not a coding RNA, the authors need to do more than bioinformatically assess using a single programme. Ideally this should be experimentally verified or the conclusions appropriately qualified.

Answer: Another bioinformatic programme named CPAT (PMID: 23335781) also predicts Trincr1 as a noncoding RNA (**Table R1**). In addition, for the 140 nt fragment (T4) that repressed ERK phosphorylation in our experiments, the longest ORF codes for a polypeptide only 6 amino acids. Finally, Ribosome-seq experiments from other studies show that Trincr1 is not translated (PMID: 22056041 and 29843593). These data support that Trincr1 is a noncoding RNA.

Table R1: Trincr1 coding potential predicted by CPAT

Data ID	Sequence Name	RNA size	ORF size	Ficket Score	Hexamer Score	Coding Probability	Coding Label
0.00	TRINCR1	502.00	132.00	0.68	-0.14	0.00	no

What is “control” in Fig. 1e? This is not mentioned in the text.

Answer: The control is ESC transfected with empty 3XFLAG overexpression vectors. We added the information in the revised version in page 28.

Legend Figure 2c. “For each gene, data were normalized to the mRNA level of mock transfected ESCs.” I thought this was endogenous levels of Trincr1.

Answer: We are sorry to make this mistake. We changed this sentence to “For each gene, data were normalized to the RNA level of wild type ESCs.” in the revised manuscript.

Legend Fig 5d not sure if they are referring to T5 instead of T6 as there is no T6 in the figure.

Answer: It is T5. We corrected this typo in the revised manuscript. Thank you!

REVIEWERS' COMMENTS:

Reviewer #1 (Remarks to the Author):

The authors have addressed all the concerns and comments raised by this reviewer, and have substantially improved the manuscript. There are no further questions.

Reviewer #2 (Remarks to the Author):

The authors have made a commendable effort to improve the quality of the manuscript and have addressed all my concerns, either by rationalising their claims or, more importantly, by increasing n numbers and data analysis/presentation. The current paper represents an informative study that will be interesting to a broad range of readers.

Reviewer #3 (Remarks to the Author):

The authors have done a good job in addressing all of my comments, and the data in the manuscript now support their conclusions. I remain concerned about the small changes in ERK activity levels observed and the big changes these elicit (especially under MEK repressed conditions), but the authors have provided reproducible data that indicates that these changes are functionally relevant. Although more mechanistic insights would have been good, the large body of work here is sufficient to warrant publication at this stage.

We thank all the reviewers for their efforts and critical thoughts during the reviewing process. We are happy to know that all reviewers are satisfied with the revised version with no further requests. We continue working on this direction and hopefully will have more progress to report in future.

REVIEWERS' COMMENTS:

Reviewer #1 (Remarks to the Author):

The authors have addressed all the concerns and comments raised by this reviewer, and have substantially improved the manuscript. There are no further questions.

Reviewer #2 (Remarks to the Author):

The authors have made a commendable effort to improve the quality of the manuscript and have addressed all my concerns, either by rationalising their claims or, more importantly, by increasing n numbers and data analysis/presentation. The current paper represents an informative study that will be interesting to a broad range of readers.

Reviewer #3 (Remarks to the Author):

The authors have done a good job in addressing all of my comments, and the data in the manuscript now support their conclusions. I remain concerned about the small changes in ERK activity levels observed and the big changes these elicit (especially under MEK repressed conditions), but the authors have provided reproducible data that indicates that these changes are functionally relevant. Although more mechanistic insights would have been good, the large body of work here is sufficient to warrant publication at this stage.